# Continuous estimation of power system inertia using convolutional neural networks

**Daniele Linaro** [1] ✉, **Federico Bizzarri** [1,2], **Davide del Giudice** [1], **Cosimo Pisani**[3], **Giorgio M. Giannuzzi**[3], **Samuele Grillo** [1] & **Angelo M. Brambilla**[1]

Inertia is a measure of a power system's capability to counteract frequency disturbances: in conventional power networks, inertia is approximately constant over time, which contributes to network stability. However, as the share of renewable energy sources increases, the inertia associated to synchronous generators declines, which may pose a threat to the overall stability. Reliably estimating the inertia of power systems dominated by inverted-connected sources has therefore become of paramount importance. We develop a framework for the continuous estimation of the inertia in an electric power system, exploiting state-of-the-art artificial intelligence techniques. We perform an in-depth investigation based on power spectra analysis and input-output correlations to explain how the artificial neural network operates in this specific realm, thus shedding light on the input features necessary for proper neural-network training. We validate our approach on a heterogeneous power network comprising synchronous generators, static compensators and converter-interfaced generation: our results highlight how different devices are characterized by distinct spectral footprints - a feature that must be taken into account by transmission system operators when performing online network stability analyses.

In recent years, the fraction of power generation capacity ascribed to renewable energy sources has been growing at a quickening pace[1]: this in turn has caused a substantial increase in the share of power sources connected to the grid by means of a power electronic interface known as an inverter, hence the name inverter-based resources (IBRs). Compared to synchronous generators, which constitute the major source of power in conventional power systems, IBRs have a fundamentally different dynamical behavior, which is expected to have significant implications for the overall dynamics and stability of the power grid[2,3].

In general, power systems are kept stable by limiting frequency excursions: a common measure of a power system's capability to counteract frequency changes is its inertia, which, in conventional power systems, is related to the kinetic energy stored in the rotating masses of synchronous generators and immediately available in case of sudden power imbalances[4] (but see ref. 5 for an investigation of the role played by generator load damping in maintaining stable

synchronization). IBR-interfaced renewable energy sources, on the other hand, typically do not provide inertia to the power network. A consequence of the increase in the penetration of IBRs has thus been a reduction in the amount of power generated by conventional power plants, which in turn has led to an overall decrease of inertia together with an increase in its variability[6]: this might hinder the ability of a power system to properly counterbalance frequency oscillations due to active power imbalances[7]. Thus, besides studying ways to make IBRs mimic the inertial response of traditional generators[8], significant research efforts have been devoted recently to the development of methods for the estimation of the inertia of a power system, some of which have been reviewed in ref. 9. These can be roughly classified into two broad categories: *(i)* algorithms triggered by a considerable disturbance (i.e., a significant event in the power system under study); *(ii)* methods that either use the measurements under normal operating conditions or rely on the transient response to probing signals injected

[1]DEIB, Politecnico di Milano, P.zza Leonardo da Vinci 32, Milano 20133, Italy. [2]ARCES, University of Bologna, Bologna 41026, Italy. [3]Terna Rete Italia S.p.A., V.le Egidio Galbani, 70, Rome 00156, Italy. ✉e-mail: daniele.linaro@polimi.it

to seamlessly stimulate the power system. The approaches in the first group analyze the measurements of electrical frequency and active powers after a significant disturbance was detected[10,11]. When they are intended for online estimation, finding the exact instant the disturbance took place is of paramount importance, as misjudgments significantly affect the estimation process. Additionally, these algorithms fail to provide updated inertia values on a continuous basis, as they need a triggering event[12,13]. Concerning the second group of methods, techniques that do need a probing signal to be fed into the power systems are impractical for large power systems, and the perturbing signal does influence the estimation[14]. On the other hand, the methodologies employing ambient measurements need to run a system identification procedure[15,16], or rely on the knowledge of accurate real-time data[17], both potential limitations to the techniques. We refer the interested reader to[18,19] for in-depth reviews on the topic of inertia estimation in power systems.

As explained in more detail in the Methods, an equivalent way of describing the inertial characteristics of a power network is by means of its momentum. Indeed, the inertia of a network comprising $N$ synchronous generators is given by $\sum_{i=1}^{N} H_i S_i / \sum_{i=1}^{N} S_i$, where $H_i$ and $S_i$ are respectively the inertia constant and apparent rated power of the $i$-th generator: this effectively coincides with the inertia of the center of inertia (COI) of the network. Momentum, on the other hand, is given by $2 \sum_{i=1}^{N} H_i S_i / f_n$, where $f_n$ is the nominal operating frequency of the network. While these two measures convey the same type of information, momentum has the advantage of being an "incremental" quantity, i.e., if a device with inertia is added to a power network, momentum will always increase. The inertia of the COI, on the other hand, could remain unchanged if the inertia of the new device were the same of the COI (i.e., the *average* inertia of the network). In order to be able to distinguish between these two scenarios—and in line with other works in the literature[20]—, in the following we will use momentum instead of inertia.

In this paper, we develop a framework for the online estimation of momentum based on a convolutional neural network (CNN)[21,22]: CNNs are a particular category of artificial neural networks (ANNs)[23]

that have become the de facto standard for classification tasks[24], particularly in the field of computer vision[25]. The inputs to the CNN are the time series of a set of electrical quantities recorded at a reduced number of buses of the network, while the output is the momentum of one or more areas of the power network. Our goal is therefore to have a CNN learn the relationship between time series of electrical variables and corresponding values of momentum. Our approach is entirely data-driven: rather than making assumptions on the underlying model of a power system (as done for instance in refs. 15,16,26–29), it uses voltage measurements at a limited number of buses during normal operation by exploiting the continuous perturbations attributable to the stochastic fluctuations of loads, i.e., of the power balance. Additionally, it provides a continuous estimation and, as such, can be used to predict in real time the momentum of a power system. Validation of this approach was performed using synthetic data generated by simulating the well-known IEEE 39-bus benchmark system modified to appropriately model the intrinsic fluctuations of the power loads in the network. By using spectral analysis of the inputs and studying the input-output correlations of the convolutional layers, we provide a systematic explanation of how the CNN works, which is crucial for instructing the amount and typology of data that should be included in the training set to achieve a desired level of performance.

## Results and discussion

We modified the IEEE 39-bus system[30] shown in Fig. 1 and used it as a benchmark to illustrate our approach to the estimation of momentum. The original network, a simplified model of the New England power system, contains 46 lines and 10 generators, with G1 modeling the aggregate behavior of a large number of generators: this is reflected in its nominal power ($S_{G1} = 10$ GVA), which is one order of magnitude larger than those of the other generators. To add to the network a device capable of providing inertia but different from the synchronous generators, we connected a synchronous compensator at bus 8. Additionally, each load in the network is stochastic, which has the goal of perturbing the network from its operating point, thus exposing the

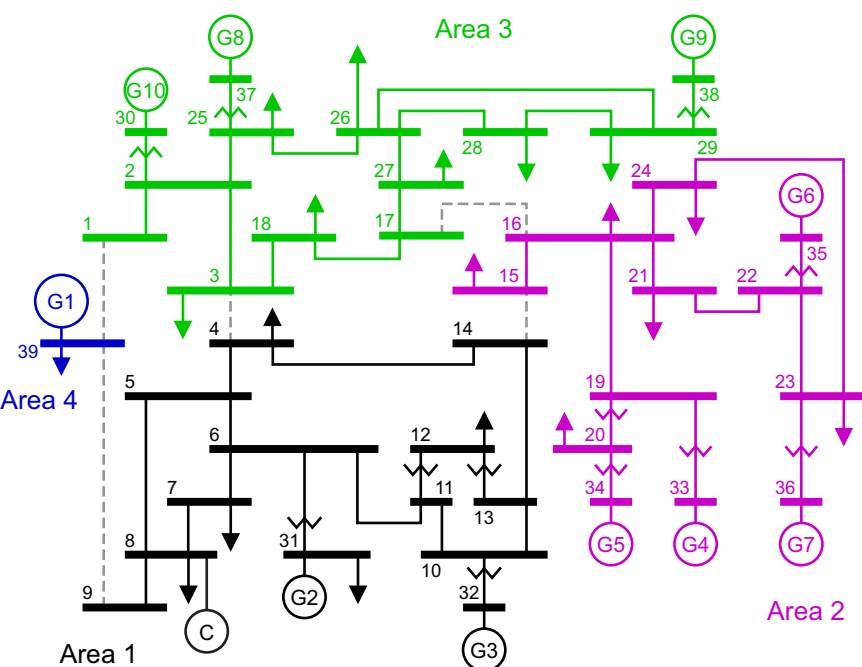

**Fig. 1 | Schematic of the IEEE 39-bus system.** Different colors highlight the areas in which the network was subdivided. The gray dashed lines are transmission lines connecting distinct areas. Area 1 contains a static compensator (labeled C) not present in the original network. Area 4 contains only the generator G1, bus 39 and the associated load.

richness of its dynamics. More details about the compensator and the loads are given in the Methods.

Unless stated otherwise, we focus on the estimation of the momentum of area 1, which contains the generators G2 and G3 and the compensator, as this provides an excellent test-bench to showcase our method. Our approach can of course be readily extended to additional areas or more complex scenarios.

## Voltage spectra upon area momentum variation

As a first step towards understanding how a machine learning (ML) model can learn to associate the dynamics of a given set of electrical quantities to specific values of momentum, we analyze the spectral properties of the voltage at a bus of the network. Here, we present results for the direct axis component of voltage at bus 3 (henceforth indicated as $V_{d,3}$), but analogous considerations are valid for the (direct and quadrature) voltages at other buses in the network. Figure 2 shows a summary of the dynamical behavior of $V_{d,3}$ for different values of momentum of area 1, obtained by varying the inertia constant of the generators G2 and G3, according to the grid displayed in Fig. 2a. The inertia constant of the compensator connected to bus 8 was fixed at 0.1 s, in order to have a negligible additional impact on the overall area momentum. The nominal values of the inertia constant of G2 and G3 are 4.33 s and 4.47 s, respectively: we therefore decided to span an interval of (−1, +1) s with respect to the nominal values for each of the two generators and sampled the inertia plane in Fig. 2a at the points indicated with white circular markers. Each of these points corresponds to a distinct value of area momentum, ranging from 0.17 to 0.27 GWs². While the effect of changing the area momentum is not evident on the example voltage traces shown in Fig. 2b (which are normalized, over the whole dataset, to have zero mean and unitary standard deviation, see Methods), it becomes more apparent when looking at the shape of the distributions of the voltage samples over longer simulation times, as shown in Fig. 2c. Indeed, the distributions' means are approximately 0 for all momentum values: this is a consequence of having subtracted, when normalizing, the voltage value of the power-flow (PF) solution, which is not affected by the inertia of the generators. The standard deviation of the distributions, on the other hand, is affected by the values of inertia, and increases with the overall area momentum, as shown in the inset of Fig. 2c. The effect of varying area momentum is even more evident when looking at the average spectra of hundreds of 60 s-long simulations (Fig. 2d, e). In these panels, the colors of the traces indicate the corresponding pair of inertia constants of G2 and G3, according to the color code shown in Fig. 2a. We see that higher values of momentum (i.e., higher inertia constants, exemplified by the orange and yellow traces) cause a shrinking of the voltage samples distributions and a shift towards lower frequencies of the peak of the voltage spectra located around 1 Hz. This same shift of the peak is evident when looking at the spectra in panel e, where the red (blue) traces correspond to higher (lower) levels of area momentum. Indeed, as will be shown in greater detail in the following, the frequency band in the range (0.5, 2) Hz is the one where changes in the inertia constant of the synchronous generators are more apparent. These spectra are a footprint of how generators, in a broad sense, contribute inertia to the power system. The frequency location and bandwidth of these magnitude peaks may also allow the identification of types of "equipment" that contribute inertia to the network, such as synchronous generators, synchronous condensers, and grid-forming converters. For instance, one can clearly see that peaks in the (4, 10) Hz frequency band in Fig. 2d, e do not change substantially when the inertia of G2 and G3 is varied: indeed, these peaks are related to inter-area oscillation modes[31] and, as will be shown in the following, are mainly due to the inertia of synchronous compensators.

## Two-value momentum estimation

As a first test of the capability of a CNN to correctly estimate the momentum of a power system, we trained a network whose task was to

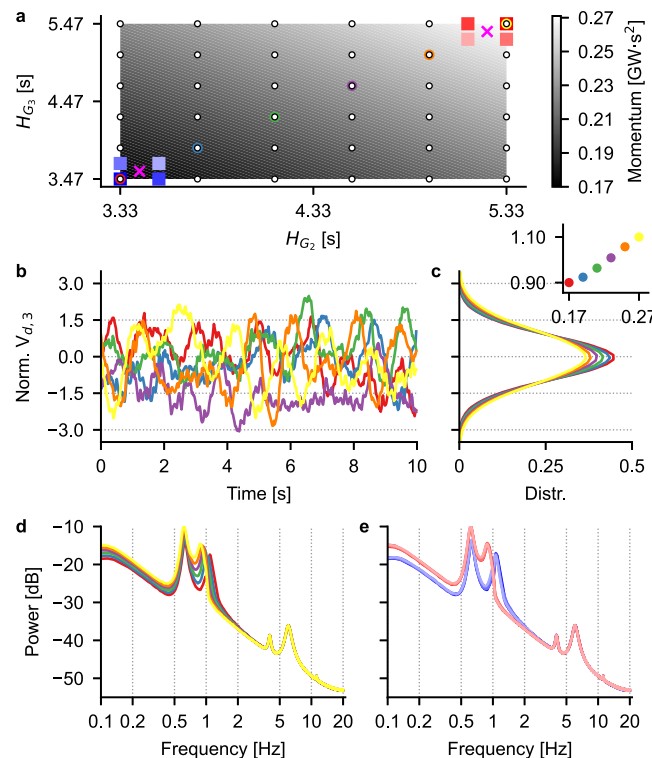

**Fig. 2 | Time- and spectral-domain analyses of $V_{d,3}$. a** Values of inertia constant of the synchronous generators in area 1 and corresponding area momentum (encoded in shades of gray according to the colorbar on the right). White circular markers indicate all the $H_{G_2}$ and $H_{G_3}$ pairs used to build the training set. Colored circular markers on the diagonal correspond to the example traces shown in the following panels, while blue and red square markers indicate additional values of momentum used for training a simpler CNN, with the magenta crosses being their average values (see main text). **b** Example normalized voltage traces for the six values of area momentum on the diagonal of the grid in **a**. **c** Distributions of the normalized voltage traces shown in **b** computed over several hundreds of 60 s-long simulations. Inset: standard deviation of the distributions as a function of area momentum. **d** Average power spectra of the voltage traces in **b**. **e** Average power spectra of the voltage traces corresponding to the points indicated with square markers in **a**.

differentiate between two well-separated values of momentum: we reasoned that by training a CNN to perform this simpler task, we would be able to gain an understanding of how the network solves this problem, and in particular which features of the input are crucial for a successful prediction. The two momentum values are the ones indicated with magenta crosses in Fig. 2a and correspond to the average momenta of the four low- (high-)momentum points indicated with blue (red) square markers in the same panel, i.e., 0.176 GWs² and 0.266 GWs². The reason for choosing four relatively close points instead of just one lies in the fact that we wanted to expose the CNN, during training, to different combinations of inertia constants of the generators G2 and G3 that lead to relatively similar values of momentum, in order to maximize the generalization capabilities of the CNN. The inertia constants and corresponding momenta used for the training set are summarized in Table S1. Figure 3a shows five normalized voltage traces for each of the low- and high-momentum condition (green and magenta traces, respectively), while the overall distributions of the training traces are shown in Fig. 3b: these display a clear signature of the effect of increasing area momentum on the voltage dynamics. This is further exemplified in the power spectra shown in Fig. 3e: as discussed earlier, the most marked differences are in the frequency range (0.5, 2) Hz. The validation and test sets were also composed of eight different combinations of inertia constant each and

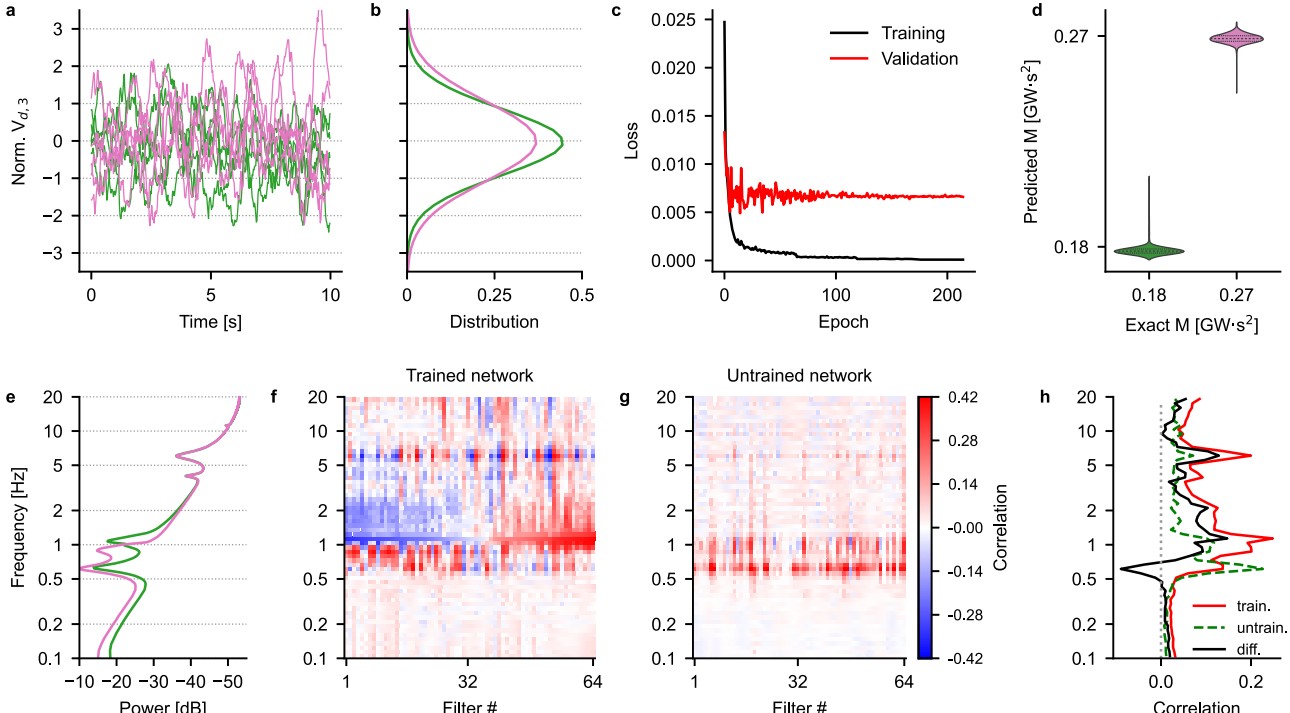

**Fig. 3 | Training a CNN for area momentum estimation.** Example traces (**a**) and corresponding distributions (**b**) for the low- and high-momentum cases (green and magenta traces, respectively). **c** Evolution of the training and validation losses as a function of the training epoch: no overfitting is apparent. **d** Violin plots of the CNN predictions on the test set data indicating a good agreement between target and predicted values. **e** Power spectral densities (PSDs) of the voltage traces showing clear differences between the momentum levels in the band $(0.5, 2)$ Hz.

**f, g** Correlation maps of the last convolutional layer for the trained (**f**) and untrained (**g**) network, sorted according to the correlation values in the 1.1 Hz band and with the frequency range subdivided into 60 logarithmically-spaced bins. See Fig. S1 for the effect on correlation magnitude of changing the number of subdivisions. **h** Mean absolute correlation computed over all filters for the trained (solid red trace) and untrained (dashed green trace) networks. The black trace is the difference between the two.

averaged to give low and high momenta: for the validation and test sets, the values of the inertia constant for the two generators were offset by 67 ms and 133 ms, respectively. In this configuration, the CNN only has one input: the direct axis component of the voltage at bus 3, i.e., $V_{d,3}$. The results of the training are shown in Fig. 3: panel c displays the evolution of the training and validation losses as a function of the training epoch, while panel d shows violin plots of the prediction of the network on the test set (mean absolute percentage error (MAPE) equal to 1.79%). These results indicate that the CNN is capable of learning the relationship between voltage dynamics and corresponding momentum. Similar results can be obtained by training a CNN using $V_{q,3}$, i.e., the quadrature axis component of the voltage at bus 3, as shown in Fig. S2b.

To comprehend the mechanism at the basis of the network's capability to correctly predict the area momentum, we performed a similar analysis as the one described in ref. 32, which consists in building so-called "input-feature unit-output correlation maps": these maps measure the correlation between the output of a given unit (a neuron) in one of the convolutional layers of the CNN and the power of the samples in the neuron's receptive field (RF), i.e., the subset of samples in the whole 60 s-long trace that affects the output of each unit in a convolutional layer (see ref. 33 for a thorough explanation of RFs and[34] for a practical implementation). This analysis is performed for different frequency bands: thus, given that changes in area momentum have a clear effect on the power spectra of the input signals (see Figs. 3e and 2d), correlation maps are a powerful tool to visualize the frequency bands the CNN is most sensitive to when predicting area momentum. Briefly, to compute a correlation map, the input signal is bandpass-filtered in one of several frequency bands in the range $(0.1, 20)$ Hz. The choice of this frequency band is dictated by the location in the frequency domain of the electro-mechanical modes

of an electrical power system that display sensitivity to the inertia of synchronous machines/motors and of the virtual inertia provided by inverter-based resources: indeed, these are all located well within the 20 Hz upper frequency limit we have chosen. For each frequency band, one computes the squared mean envelope for each receptive field in a given layer and then calculates the correlation between the envelope and the output of the same layer in response to the unfiltered input signal. For a more detailed description of how to compute correlation maps, the reader is referred to ref. 32. The results of this analysis are summarized in Fig. 3f-h: panel f shows the correlation measured in the trained network as a function of the frequency for each of the 64 filters of the last convolutional layer before the dense layer, sorted according to the correlation values at a frequency of 1.1 Hz. High correlation values can be observed in two non-overlapping bands: the first one covers approximately the range $(0.5, 1)$ Hz, while the second one covers the range $(1, 3)$ Hz. However, high correlation values in the former range are likely to be a by-product of the fact that the signal is stronger in that frequency band: this is confirmed by almost equally high correlation values in the map obtained with the untrained network (i.e., a network with the same architecture, but with randomly set weights) shown in Fig. 3g. This is further confirmed by panel h, which shows the mean absolute value of correlation over all the filters as a function of the frequency, for the trained (solid red line) and untrained (dashed green line) networks, together with their difference (solid black line): this last trace, in particular, shows a major correlation peak around approximately 1.2 Hz, corresponding to the presence of the peak in the spectra of the low-momentum traces (green trace in Fig. 3e). These results suggest that the features of the input signals that matter the most for momentum prediction lie in the frequency range that starts just below ∼1 Hz and extends up to ∼3 Hz. As shown in Fig. S2, this correlation analysis was carried out on CNNs trained to

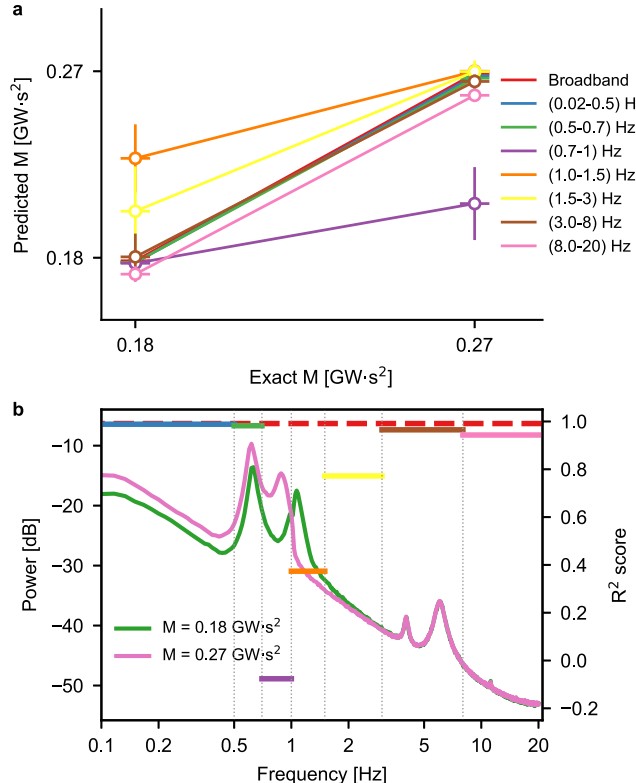

**Fig. 4 | Effect of removing selected frequency bands of the input on the accuracy of the CNN prediction. a** Predicted momentum values when non-overlapping frequency bands were removed from the input voltage traces (error bars: mean ± standard deviation). **b** $R^2$ score between the predicted momentum values obtained when no frequency was removed from the input and those obtained when each frequency band was filtered out, superimposed to the average PSDs of the low- and high-momentum conditions.

predict the momentum values of either area 1 or area 2 using either the direct or quadrature axis components of the voltages at bus 3. In all cases, the CNN was capable of correctly predicting the momentum— albeit with significantly higher MAPEs in the case of area 2, see Fig. S2c, d−, and with correlation maps characterized by strikingly similar structures in all cases.

To further validate our hypothesis that distinct frequency bands contribute differentially to momentum prediction, we filtered the input voltage traces with band-stop filters that selectively removed non-overlapping frequency bands covering the range (1, 20) Hz. These filtered traces were then fed to the CNN and the accuracy of the prediction was compared to that obtained with the original unfiltered traces, with the aim of establishing which frequency band has the highest impact on the output of the CNN. The results of this experiment are shown in Fig. 4: the top panel contains a summary of the accuracy of the prediction for each of the removed frequencies. In most cases, the prediction is very close to the one obtained with the unfiltered (broadband) signal, except for the frequency bands (0.7, 1) Hz, (1, 1.5) Hz, and (1.5, 3) Hz. Removal of the first band from the input traces causes a worsening of the prediction for high values of momentum (purple markers and error bars, indicating mean and standard error of the mean (SEM) of the predicted values, respectively), while removing the last two causes a worsening of the prediction at low values of momentum (orange and yellow markers and error bars). The bottom panel shows, for each frequency band removed from the input traces, the corresponding $R^2$ score, i.e., the agreement between the prediction in the stopband case and that of the broadband signal: perfect agreement would correspond to an $R^2$ of 1, while

lower values indicate progressively worse predictions. The $R^2$ scores are superimposed to the average power spectra corresponding to the low and high levels of momentum and clearly indicate that the most important frequency bands for an accurate prediction are those in the ranges (0.7, 1) Hz, (1, 1.5) Hz, and (1.5, 3) Hz with the former playing the most crucial role, in agreement with the results shown in Fig. 3.

Taken together, these results indicate that a CNN trained on voltage traces recorded at one bus is capable of correctly estimating the area momentum, and it does so by tuning the filters in its preprocessing pipeline to "emphasize" those frequency bands of the input signals that convey the most information about the momentum of a given area of a power network.

## Momentum estimation with added compensators
So far, area momentum was varied by changing the inertia constant of the synchronous generators G2 and G3. However, as mentioned previously, a compensator was connected to bus 8 of the IEEE 39-bus network (see Fig. 1) with the aim of having an additional device capable of adding momentum to area 1. In the simulations described so far, the inertia constant of this compensator was set to 0.1 s, thus making its contribution to area momentum negligible.

The peculiarity of compensators is that they can induce inter-area oscillations in a power network that are reflected in peaks in the power spectral density (PSD) around 5 Hz, i.e., in a frequency range that is not used by the CNN trained earlier to predict the momentum: we therefore expect the CNN to make large prediction errors when the area momentum is varied by acting on the compensator's inertia constant rather than on the synchronous generators' one. To directly test this, we ran simulations with the parameters listed in Table S2: the lowest and highest area momenta (first and fourth row) correspond to the values used for the test set and serve as a "control" group, while the inertia constants in the second and third row lead to the same value of area momentum by increasing either the inertia constants of G2 and G3 (second row) or the inertia of the compensator from 0.1 s to 6.1 s (third row). This relatively higher increase is due to the fact that the rated power of the compensator (100 MVA) is significantly lower than that of either G2 and G3 (700 MVA and 800 MVA, respectively). The PSDs corresponding to these four conditions are shown in the top part of Fig. 5a: blue and orange traces are the low and high momenta used for the test set, respectively, while the green (magenta) trace corresponds to a momentum of 0.197 GWs² with low (high) compensator's inertia constant. The shift in the peak around 5 Hz and the separation of the spectra at frequencies above ~6 Hz due to the increase in compensator's inertia are evident in the magenta trace, while the other three traces overlap in that frequency range. The bottom part of Fig. 5a shows enlarged versions of the PSDs: in the (0.4, 1.5) Hz range, the magenta and blue traces effectively overlap, since the inertia constants of G2 and G3 in these two conditions are very similar. In the (8, 15) Hz range, the blue, orange and green traces are identical, while the magenta one shows a prominent peak around 11 Hz and has an overall lower power spectrum.

As expected, a CNN trained without variable compensator inertia can correctly estimate the area momentum when the inertia constants of G2 and G3 are varied (green square marker in Fig. 5b), but not when the inertia of the compensator is set to 6.1 s: indeed, in this latter case the prediction of the CNN is 0.184 ± 0.003 GWs² (mean ± SEM, magenta square marker in Fig. 5b) when the actual momentum is 0.197 GWs², as detailed in Table S2. To account for the presence of a compensator in area 1, we therefore expanded the training set to include a condition in which the inertia constant of the compensator was increased to 5 s: in other words, we added to the grid shown in Fig. 2a an additional "dimension" (the third inertia constant), thus effectively doubling the amount of data used for training the CNN. The training with this increased amount of data evolved in a similar fashion to that shown in Fig. 2c and the corresponding CNN had a MAPE on the

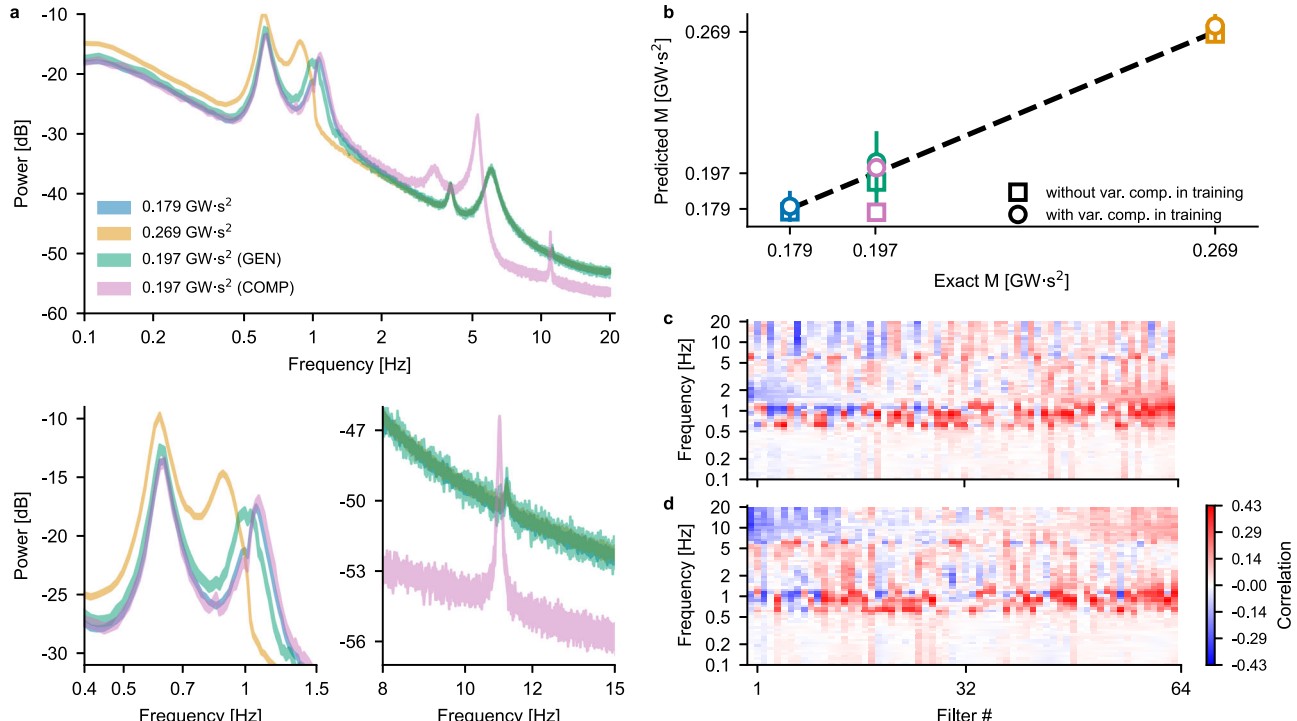

**Fig. 5 | Momentum prediction when a compensator's inertia constant is varied.**
**a** Top, PSDs of the voltage traces in the various conditions outlined in Table S2.
Notice how up until -1.5 Hz the magenta and blue traces overlap, while above this
frequency value the blue trace coincides with the green one. Bottom, enlarged
views of the PSDs in two frequency bands important for the CNN operation.

**b** Square (circular) markers indicate the momentum values predicted by a CNN
trained without (with) variable compensator data. Marker colors correspond to the
conditions shown in **a** (error bars: mean ± standard deviation). **c**, **d** Correlation
maps sorted according to the correlation values in the 1.1 Hz (**c**) and 10 Hz fre-
quency bands (**d**).

test set of 0.87%, indicating once again that a convolutional neural
network is a suitable tool for learning the relationship between net-
work dynamics and corresponding momentum. Additionally, this
second CNN is capable of correctly predicting the momentum in both
conditions corresponding to an area momentum of 0.197 GWs$^2$, as
detailed in Table S2: when the inertia of G2 and G3 is varied, while
leaving that of the compensator equal to 0.1 s, the prediction of the
CNN is $0.192 \pm 0.013$ GWs$^2$ (magenta circular marker in Fig. 5b); on the
other hand, when the compensator's inertia is increased to 6 s, the
prediction is $0.197 \pm 0.002$ GWs$^2$ (green circular marker in Fig. 5b),
much closer to the real value than what was achieved with the
first CNN.

To better understand the changes in the "tuning" of the filters that
make up the preprocessing part of the CNN, we resorted once again to
the correlation analysis introduced earlier. The results for the CNN
trained on the dataset including the variable compensator data are
shown in Fig. 5c, d (filters sorted according to the correlation values in
the frequency band around 1.1 Hz and 10 Hz, respectively). These
correlation maps highlight how the CNN is sensitive not only to the
(1.5, 3) Hz frequency range, but also to frequencies above approxi-
mately 7 Hz, which indeed correspond to a range where the presence
of the compensator causes a significant downward shift of the spec-
trum. As discussed for Fig. 3, high correlation values in the range (0.5,
1) Hz are due to the strong signal components that are present in the
spectra for all values of momentum: these reliably drive the output of
the preprocessing pipelines even in the case of the untrained network
(see Fig. 3g) and are therefore not used by the CNN to perform the
classification. Overall, these results indicate that, in order to achieve as
accurate a prediction as possible, a CNN should be trained with data
that cover as many "spectral conditions" as possible, as this is neces-
sary to have an adequate tuning of the filters that constitute the pre-
processing pipeline of the network.

## Continuous momentum prediction

So far, in order to gain a mechanistic understanding of how a CNN can
learn to predict the value of area momentum, we have considered
networks trained on a limited subset of the data shown in Fig. 2. In
order to extend our approach, we used the full dataset (i.e., the grid of
points shown in Fig. 2a) to train a CNN capable of generating a con-
tinuous estimation of area momentum in the range (0.17, 0.28) GWs$^2$.
We included in the training dataset not only the direct voltage recor-
ded at bus 3, but also the ones recorded at buses 14, 17 and 39. While it
is still possible to use only one voltage and train sufficiently accurate
CNNs, the accuracy of the prediction increases substantially when
using multiple voltage traces, while not compromising the practical
feasibility of this choice. Figure 6 shows the evolution of the loss
function during training (top panel) and the performance of the CNN
on the test set (bottom panel). As it can be seen, the network does not
overfit and learns to accurately predict the area momentum (MAPE on
the test set: 2.67%). Interestingly, the performance of the CNN is
slightly lower only in the center of the momentum range: this is
probably due to the fact that several combinations of generators'
inertia can lead to the same values of momentum in the range (0.21,
0.23) GWs$^2$, thus testing the generalization capabilities of the network.

To probe the extent to which this network can predict stepwise
changes in area momentum, we performed the experiments shown in
Fig. 7, consisting of four different conditions (one for each panel):
(A)  area momentum changed by changing the inertia of the area
     generators;
(B)  area momentum constant while changing the inertia of the area
     generators;
(C)  area momentum increased by increasing the inertia of the area
     generators;
(D)  area momentum increased by the same values as in (C) but with
     increases in the inertia of the area compensator.

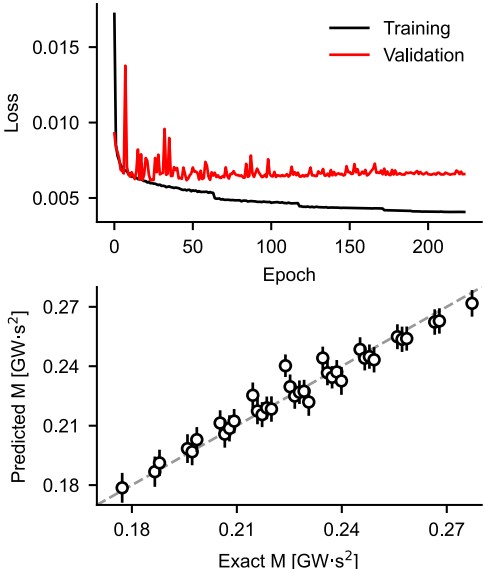

**Fig. 6 | Training a CNN to predict continuous values of area momentum.** Top panel: training and validation loss as a function of epoch number. Bottom panel: mean and standard deviation (circular markers and error bars, respectively) of the CNN predictions on the test set. The number of distinct values of momentum to predict is equal to the number of dots in the grid of Fig. 2a and adds up to 36.

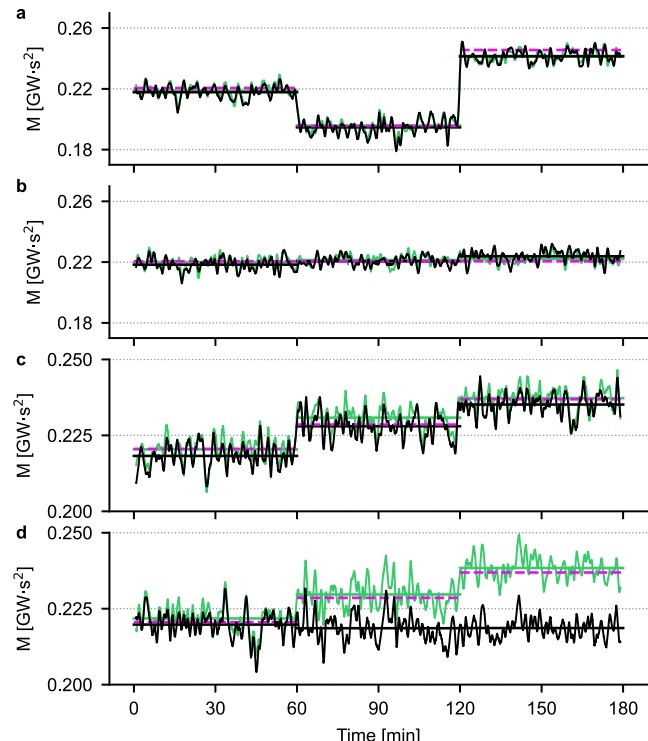

**Fig. 7 | Area momentum prediction upon step-wise inertia variations.** In all panels, the black trace is the value of momentum predicted by a CNN trained on a dataset where the compensator's inertia was fixed at 0.1 s, while the green traces are the predictions of a CNN trained on an extended dataset in which, for each pair of inertia values shown in the grid in Fig. 2a, three values of compensator's inertia were considered, namely 0.1 s, 2.5 s and 5 s. **a** Different values of momentum obtained by changing the inertia of G2 and G3. **b** Fixed value of momentum obtained for different combinations of the inertia of G2 and G3. **c** Different values of momentum obtained by progressively increasing the inertia of G2 and G3. **d** Same values of momentum as in **c**, obtained by increasing the inertia of the compensator in area 1.

The exact values of inertia of the generators G2 and G3 and of the compensator in area 1 are reported in Table S3. For each of the four conditions, a 3 hour-long simulation was performed, during which the synchronous generators' or the compensator's inertia was changed twice, leading to three 1 hour-long intervals at constant momentum. The traces in Fig. 7 are a moving average (in steps of 1 s) of the predictions of the CNN: as it can be seen, the quality of the prediction is excellent in all conditions, except for panel d, where the area momentum is increased by raising the compensator's inertia from 0.1 s (lowest value of momentum, $0.2206\,\mathrm{GWs}^2$) to 2.5 s and 5 s (corresponding to momentum values of 0.2286 and $0.2369\,\mathrm{GWs}^2$, respectively). The reason for this failure lies in the fact that varying the compensator's inertia changes the spectra of the voltage traces in a frequency range that is overlooked by the CNN when predicting the momentum, as discussed earlier for the simpler case of low and high values of momentum (see Fig. 5). To solve this problem, we augmented the training set by including two additional values of compensator's inertia, namely 2.5 s and 5 s: this effectively tripled the amount of data used in the training, as the grid shown in Fig. 2a was replicated for each of the two additional values of compensator's inertia. As expected, a CNN trained on this larger dataset (MAPE on the test set: 2.24%) is capable of correctly predicting changes in area momentum even when only the inertia of the compensator is increased (green traces in Fig. 7).

As previously, we resort to spectral analysis of the voltage traces to justify the necessity to include in the training set additional simulations at varying values of compensator inertia. The results of this analysis are shown in Fig. 8: panel a contains representative spectra for different values of generators' and compensator's inertia. For each value of compensator's inertia, the red traces are the spectra of the voltage traces at lower momentum (i.e., 0.17, 0.18 and $0.19\,\mathrm{GWs}^2$ in panel b), that are obtained when the inertia of each generator is set at its lowest value. Increasing the generators' inertia causes a shift in the peaks of the spectra in the range (0.5, 1.5) Hz, thus leading to the blue traces (highest values of momentum, 0.27, 0.28 and $0.29\,\mathrm{GWs}^2$ in panel b). On the other hand, increasing the compensator's inertia from 1 s to 6 s causes a leftward shift (i.e., towards lower frequency values, as exemplified by the arrow in panel a) of the peak in the spectra located

between 5 and 20 Hz (different shades of the red and blue traces). Figure 8b shows spectrograms obtained for three different values of compensator's inertia (i.e., 1, 3 and 6 s). In these panels, each row corresponds to a PSD like the ones shown in panel a, with warmer (cooler) colors indicating higher (lower) values of the PSD. The inertia of the compensator is fixed at the value indicated in the upper right corner of each panel, while the area momentum is varied by changing the inertia of the generators G2 and G3. This results in a leftward shift of the second PSD peak as the momentum is increased, as shown by the blued dashed line. The white dashed line shows the location of the first significant peak of the PSD, which remains unchanged despite changes in generators' inertia. The white arrowhead indicates the location of the high-frequency peak modulated by the value of inertia of the compensator. Figure 8b clearly highlights how the low- and high-frequency peaks are differentially modulated by changing either the generators' or the compensator's inertia. Finally, Fig. 8c shows the location of the high-frequency peak as the compensator's inertia is increased: the monotonicity of this curve allows the CNN to correctly learn the relationship between voltage spectra and momentum. The spectra in Fig. 8 are from the direct voltage traces recorded at bus 3, but analogous considerations are valid for the other voltages used to train the CNNs used in this section.

These results once more highlight the capability of a CNN to correctly predict the area momentum in several operating scenarios, assuming that the network has been trained on an appropriately constructed dataset. In particular, having shown that the convolutional part of the CNN performs a linear filtering of the input traces that

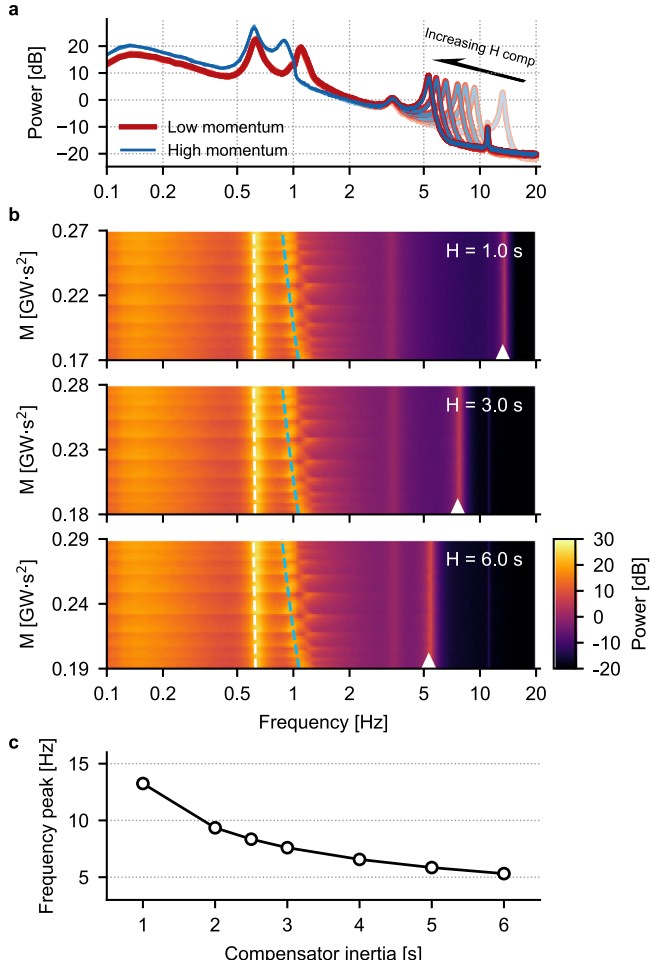

**Fig. 8 | Spectral analysis of the data used for training the CNN. a** Example PSDs for different momentum values: see the text for an explanation of the color code. **b** Spectrograms in the (0.1, 20) Hz frequency range for different values of momentum and for the three values of compensator's inertia. The white and blue dashed lines indicate the position of the two low-frequency peaks of the PSDs and are fit of the actual positions with functions of the form $y = ax^b$, with $a$ and $b$ parameters fit to the data. White arrowheads are placed at the location of the high-frequency peak of the PSDs. **c** The location of the high-frequency peak displays a monotonous dependence on the compensator's inertia.

preserves the most information-rich parts of the spectra, failure to include in the training set operating conditions that activate specific frequency bands will result in incorrect predictions when a significant component of the spectrum is indeed present in such frequency bands.

We have presented a CNN-based approach for continuously estimating the momentum of a power system, by framing the estimation problem as a classification task in which the inputs to the CNN are the time series of a set of electrical quantities recorded at a reduced number of buses, while the output is the momentum of one or more areas of the power network. The CNN architecture used here was inspired by[35] and modified to take into account the peculiarities of power systems' data. We trained CNNs that, given 60 s of voltage at a limited number of buses (four at most), can estimate the momentum of an area of a power network. We have shown that the weights of the convolutional layers are tuned to exploit peculiar spectral features of the dynamics of the power system in order to extract the momentum values. This is a relevant aspect since it is oftentimes difficult to obtain a complete understanding of the mechanisms underlying the functioning of a CNN. In the test-cases considered in the present study, the

MAPE on the prediction rarely exceeded 4%, indicating that CNNs can be successfully employed in this type of tasks. The advantage of this approach over more conventional inertia estimation algorithms is that it provides a continuous prediction and hence does not require network events to update its output. In particular, our method is capable of rapidly detecting changes in area momentum (as shown in Fig. 7) and can be used when these are attributable to both changes in generators' and/or compensators' inertia (see Fig. 8).

By taking a reductionist approach, we have shown that the mechanism at the basis of the functioning of the CNN consists in a linear filtering of the inputs by the convolutional preprocessing part that preserves the most salient spectral features of the input traces (see Figs. 3 and 4), which can then be efficiently classified by the dense part of the CNN. The major advantage in using a CNN to perform this task lies in the fact that the frequency bands that carry the most information are determined by the optimization algorithm during training, rather than having to be selected "by hand".

### Impact of additional devices on momentum estimation

A direct consequence of this data-driven approach, however, is that the addition to the power network of a device that was not present during training gives no guarantee that the CNN will be able to predict the area momentum correctly. To illustrate this point, we replaced the synchronous generator G3 with a virtual synchronous generator (VSG), i.e., a device that emulates the mechanical and partially the electrical properties of a synchronous generator, thus enabling IBR-based resources to mimic, among other things, the inertial characteristics of synchronous generators[36]. As in the previous experiments shown in Fig. 7, we tested three different values of area momentum, and switched between them by instantaneously changing the inertia of both the generator G2 and the VSG at $t = 60$ and $120$ min. Figure 9a shows the average PSDs of the voltage at bus 3 for each momentum value in the presence of the VSG (black traces) compared to the normal configuration of the network (i.e., with G2 and G3, red traces): the presence of a VSG significantly alters the spectrum in two frequency bands, around 1.5 Hz and 5 Hz. In particular, the location of the 1.5 Hz peak varies with momentum, while the other two low-frequency peaks located at ~0.6 Hz and ~1 Hz, which are also present in the red traces, do not change position. As a consequence, the CNN is capable of correctly predicting the value of area momentum only when the red and black PSDs overlap for frequencies < 1 Hz, as shown in Fig. 9b, where the dashed magenta traces are the correct values of momentum and the black trace is the moving average of the CNN prediction. However, what this simple example clearly shows is that different devices will typically add their characteristic "signature" to the spectrum, therefore making our method applicable to other network configurations and power devices, provided that the user performs a preliminary assessment of the effect(s) of changing a device's parameters on the spectral contents of the signal(s) used for estimating the area momentum.

### Robustness to load damping variability

An important parameter affecting system stability in power networks is load damping[37]: even though it appears in the swing equation that models the behavior of synchronous machines, damping cannot be changed freely by transmission system operators (TSOs), while at the same time being difficult to estimate in real-world scenarios. Therefore, it is important to ascertain whether a CNN trained on a dataset generated with a certain amount of load damping is robust, in the prediction phase, to variations in its actual value. All generators in the IEEE 39-bus network have a default value of load damping equal to 0: we varied this parameter in the range $D = (0, 4)$ while keeping the inertia of the generators fixed and tested whether a previously-trained CNN (i.e., one trained on a dataset generated with $D = 0$) would correctly predict the value of area momentum. As shown in Fig. 10a, the MAPE of the prediction is only slightly affected by changes in load

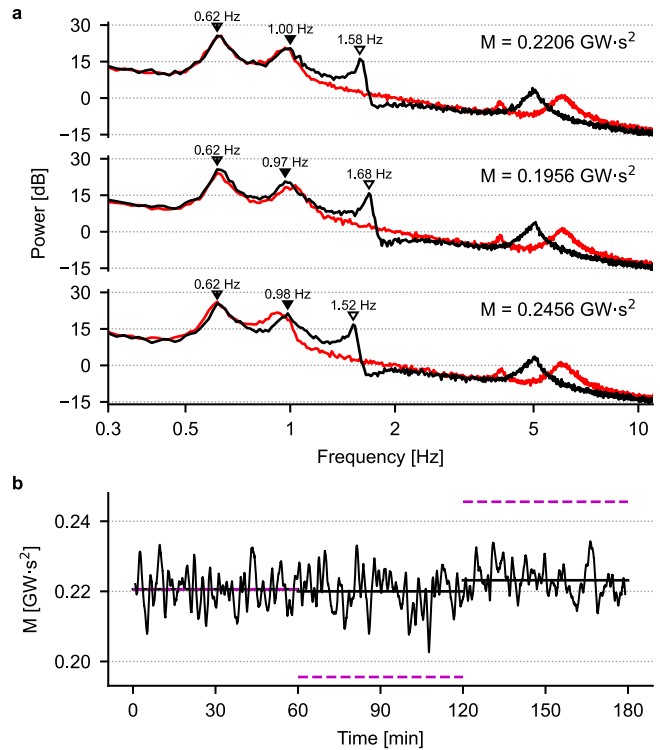

**Fig. 9 | Momentum prediction in the presence of a VSG. a** Black traces, PSDs of the voltage at bus 3 of the IEEE 39-bus network when G3 is replaced by a VSG. Red traces, PSDs of the same signal in the default power network, i.e., with the synchronous generator G3 present. The momentum values indicated in each panel have been obtained by changing either the inertia of the generator G2 and of the VSG (black traces) or the inertia of the generators G2 and G3 (red traces). Empty (filled) arrowheads indicate the positions of PSD peaks in the black traces that (do not) change position when area momentum is varied. **b** Momentum prediction by the CNN: the black trace is the moving average of the predicted momentum, while the magenta dashed lines indicate the correct value of area momentum. The prediction is accurate only when the black and red traces in panel a overlap for frequencies up to ~1 Hz.

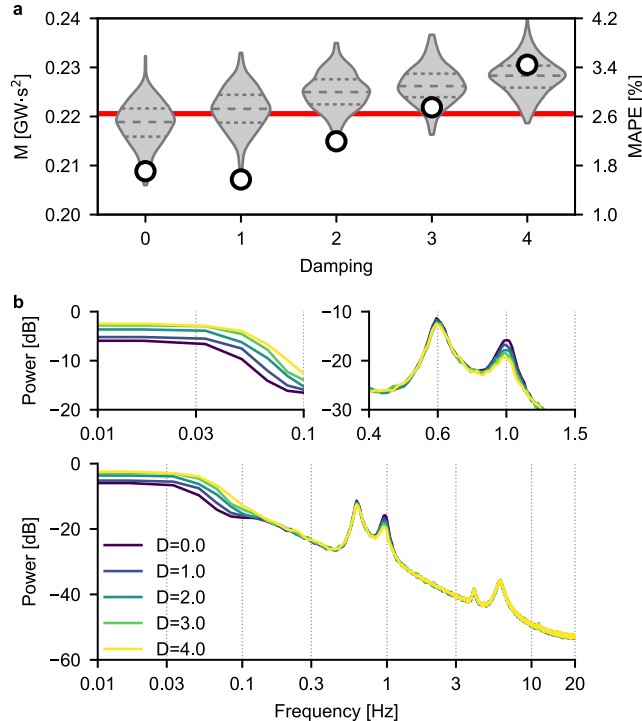

**Fig. 10 | Load damping impact on CNN prediction accuracy and voltage spectra. a** Violin plots of the predicted values of momentum when the load damping of generators G2 and G3 is increased from 0 to 4. Each violin represents the distribution of $N = 300$ predicted momentum values, with the inner dashed lines indicating, from bottom to top, the 25th, 50th and 75th percentile, respectively. The red line represents the correct value of momentum, obtained with values of inertia of G2 and G3 equal to 4.33 s and 4.47 s, respectively. The circular markers are the values of MAPE, indicated on the right axis. **b** PSDs of $V_{d,bus_3}$ for different values of load damping of G2 and G3. Insets show the PSD in those regions where the effect of varying generators load damping is more prominent.

damping, which indicates that a CNN trained with a specific (or for that matter, unknown) value of load damping is robust to changes in its value that are well within the range of what one would expect to have in a real power network. Once again, this can be explained by looking at the PSDs shown in Fig. 10b: these clearly show that the effect of load damping is either limited to very low frequencies (i.e., up to approximately 0.1 Hz) or consists in modulating the amplitude of one of the peaks of the PSD. Given that, as we have shown before, the CNN mainly relies on the location of the peaks, rather than on their amplitude, uncertainties in load damping are not expected to negatively affect the estimation of area momentum.

### Comparison with other ML methods

Previous research has investigated the application of ANNs[38,39] in general and CNNs in particular[40] to the problem of inertia estimation. While similar in the overall approach and accuracy of the prediction, our method presents two clear advantages: first, it is entirely data-driven, as it relies exclusively on the voltage fluctuations attributable to the stochasticity of power loads. This is not the case with other methods, such as[40], which instead require probing signals to perturb the system from its steady state operation point. Secondly, for the first time we provide an in-depth analysis of the mechanisms underlying the functioning of the CNN, thus providing crucial guidelines for the choice of the most appropriate training set to achieve a desired level of accuracy in the prediction of network momentum.

While the approaches just mentioned focus specifically on the estimation of network inertia, several "general purpose" ML algorithms can be used for tackling regression problems: among these, we chose multi-layer perceptron (MLP), support vector regression (SVR), kernel ridge, K-nearest neighbor and random forest for a direct comparison in terms of accuracy with our CNN-based approach. Other methods, such as linear regression, either gave unsatisfactory results or required, in our hands, excessive additional tuning. For this comparison, we used as training data the normalized $V_{d,3}$ of the low- and high-momentum dataset (see Fig. 3): importantly, in order to make the comparison as fair as possible, all algorithms were trained with exactly the same input data used for the CNN and, when possible, we (approximately) matched the number of parameters of the model with those of the CNN. The results of this analysis are presented in Table S4: as it can be seen, the approach based on CNN is superior to all other tested models. The most obvious reason behind the worse performance of these models might reside in the fact that they are not specifically intended to handle time series data: preprocessing the input data, by applying for instance Fourier and/or dimensionality-reduction techniques, might improve their performance, but was outside the scope of this work.

### Outlook

As mentioned previously, our method employs voltage data from a limited number of buses of the network: investigating in detail heuristics for the choice of bus and electrical variables that will give the best prediction accuracy is outside the scope of this work as it ultimately depends on the power network topology and its subdivision in areas: nonetheless, recent findings on the theoretically-expected statistics of

each electrical variable[41] will allow choosing the best set of variables to use during the training and prediction phases.

Additionally, in this work we have not taken into account any daily or seasonal variations in the stochastic loads present in the network. As these might play an important role in determining the overall level of momentum present in a power network, future work will be focused on modeling these aspects more accurately in order to gain a better understanding of the potential changes to our method required to make it applicable to a broader range of operating conditions. This notwithstanding, we believe that our method is robust and flexible enough to be employed with success in a vast number of power network configurations and has broad applicability to real-world scenarios.

## Methods

### Theoretical bases
The kinetic energy stored in a synchronous generator can be expressed as $E_{\text{kin}} = \frac{1}{2}J\omega_n^2$, where $J$ is the moment of inertia of the generator and $\omega_n$ is the rated angular frequency of the rotor. Starting from this equation, one can define several quantities that measure different characteristics of a synchronous generator. The first one we consider here is the inertia constant, which is defined as $H = E_{\text{kin}}/S$, where $S$ is the rated power of the generator. $H$ is measured in seconds and gives an indication of the time over which a synchronous generator is capable of providing/absorbing its rated power solely using the kinetic energy of its rotating masses[17,42].

A simplified but accurate description of the electro-mechanical dynamics of a one-mass generator is given by the swing equation[43]

$$\frac{d}{dt}\left(\frac{f}{f_n}\right) = \frac{P_m - P_e}{2HS} - D\frac{\Delta f}{f_n} = \frac{P_m - P_e}{2E_{\text{kin}}} - D\frac{\Delta f}{f_n}, \tag{1}$$

where $f$ and $f_n$ are the actual and rated frequencies of the generator, $P_m$ and $P_e$ are its mechanical and electrical powers, respectively, $\Delta f = f - f_n$, and $D$ is the load damping coefficient. Equation (1) clearly shows that the inertia constant gives a measure of how fluctuations in power lead to frequency changes: a large inertia constant (or, analogously, a large $E_{\text{kin}}$) allows a synchronous generator to effectively limit frequency fluctuations immediately after electrical power imbalances (mechanical power is usually assumed to be constant in the short term). Equation (1) can be re-written as

$$\frac{df}{dt} = \frac{P_m - P_e}{M} - D\Delta f, \tag{2}$$

where $M = 2HS/f_n$ is related to the angular momentum of the generator (to be more accurate, in classical mechanics the angular momentum is equal to $L = \frac{1}{2}J\omega = M/(4\pi)$, and thus its unit of measure is the same of $M$, i.e., $\text{kg} \cdot \text{m}^2 \cdot \text{s}^{-1}$ or $\text{W} \cdot \text{s}^2$). These three quantities, $H$, $M$ and $E_{\text{kin}} = HS$ (which has the units of measure of an energy, i.e., $\text{kg} \cdot \text{m}^2 \cdot \text{s}^{-2}$ or $\text{W} \cdot \text{s}$), are related to each other and fundamentally provide the same information: therefore, they can be used (almost) interchangeably when it comes to describing the dynamics of a power system, and in particular its ability to counteract the inevitable power imbalances that will occur in the network. Following what was done in[20], we decided to use in this work the momentum $M$ to describe a power network.

### Subdivision of a power network in areas
Our approach for estimating the momentum of a power network relies on subdividing it in a finite number of areas, as previously done in other works[11,20]. The reason for this is twofold: first, often power networks can be "naturally" subdivided in tightly connected sub-areas that are loosely interconnected with the rest of the network. Secondly, combining several synchronous generators together reduces the number of predictions that the CNN has to make, thus increasing its

capability to accurately learn the relationship between system dynamics and momentum. The momentum $M_i$ of the $i$-th area is given by

$$M_i = \frac{2}{f_n}\sum_{j=1}^{N_i} H_j S_j, \tag{3}$$

where $N_i$ is the number of synchronous generators in area $i$, and $H_j$ and $S_j$ are the inertia constants and rated powers, respectively, of the $j-$th generator in area $i$. Unlike the approach proposed in ref. 20, we do not impose any constraint on how sub-areas are defined: such subdivision should be determined on a case-by-case basis and, if one is interested in the momentum of a specific synchronous generator, they can "collapse" an area in such a way that its momentum will coincide with that of the generator in question. Indeed, this is what is done in our subdivision of the IEEE 39-bus network shown in Fig. 1, in which area 4 contains only the generator G1: in our specific case, this was done because we assume area 4 to have a known constant momentum.

### Synchronous compensator
A synchronous compensator with a nominal power $S_C = 100$ MVA was connected to bus 8 of the IEEE 39-bus network. Its active power was set to 0 MW and its voltage set point was chosen such that its reactive power is null at PF, thus not altering the operating point of the network but contributing the desired level of inertia. To find the appropriate set point, before each time-domain simulation an optimization was performed to minimize the reactive power absorbed by the compensator, which is given by

$$Q(v_g) = \text{Im}\left\{(V_d + iV_q) \cdot \overline{(I_d + iI_q)}\right\}, \tag{4}$$

where $V_d$ ($I_d$) and $V_q$ ($I_q$) are the direct and quadrature components of the compensator's voltage (current), respectively, and the bar indicates the complex conjugate. $v_g$ sets the operating point of the compensator as a fraction of $S_C$. The optimization was performed using the function fsolve of SciPy[44].

### Virtual synchronous generator
A virtual synchronous generator (VSG) is a grid-forming control scheme that simulates the dynamical behavior of a synchronous machine by means of a power converter. The goal is to provide inertia, damping, primary frequency control and voltage control to a network with a significant penetration of renewable energy sources and therefore reduced inertia. For the experiments shown in Fig. 9, we used the model of VSG described in ref. 45: briefly, it provides virtual inertia by implementing the swing equation with frequency droop control and contains a phase-locked loop (PLL) and several control algorithms that together replicate the dynamical behavior of a conventional synchronous machine. Importantly, the vast majority of VSGs that provide synthetic inertia reproduce exclusively the mechanical behavior of a synchronous machine (i.e., the swing equation), while failing to replicate its full electro-mechanical behavior due to windings, including dampers. The VSG model in ref. 45 belongs to this category and therefore implements only the swing equation, leading to a different spectral footprint with respect to a real synchronous generator.

### Stochastic loads
All the loads in our version of the IEEE 39-bus network are stochastic: the active power absorbed by each load is described by an Ornstein-Uhlenbeck (OU) process with mean equal to the value of the original load, standard deviation equal to 0.5% of the mean and a rate of mean reversion of 2 s. We chose OU processes because they have a power spectrum that is a more accurate model of the stochastic variability of power loads than Gaussian white noise[46,47]. The variance of the OU

processes is sufficiently low so that the fluctuations of the stochastic loads can be viewed as a small-signal w.r.t. the mean value of the latter. Hence, it is possible to relate the stochastic variations of all the variables of the power-system model to the fluctuations of the stochastic loads by exploiting the linearization of the power-system model at its PF solution[26,48]. In particular, the linearized power-system model, augmented by the stochastic differential equations (SDEs) that generate the OU processes from a proper set of uncorrelated Wiener processes, gives rise to a set of SDEs which is linear in narrow sense[49]. Under this hypothesis, each state variable of the power-system model turns out to be characterized by a Gaussian distribution. The asymptotic mean of this distribution is given by the value assumed by the state variable at the PF solution. The variance, in turn, depends on the variance of the OU processes. This holds also for the algebraic variables of the power-system model, i.e., those variables that are not state variables[41]. In particular, it holds for all the bus voltages, thus highlighting their relationship with the stochastic fluctuation of the loads.

## Convolutional neural network architecture

Artificial neural networks (ANNs) are a class of machine learning algorithms[23] that employ the *supervised learning* paradigm to learn the mapping relationship between input and output values using a large number of example pairs. The basic building block of ANNs is the *perceptron*[50], which loosely models real neurons by transforming a vector of (real) inputs $\mathbf{x}$ into a scalar output $y = f(\mathbf{w}^T\mathbf{x} + b)$, where $\mathbf{w}$ and $b$ are the so-called *weights* and *bias* of the perceptron and $f$ is a nonlinear *activation* function. Although limited in the range of tasks that it can solve, the perceptron spurred the development of ANNs composed of several interconnected layers capable of learning any function with an arbitrary degree of precision[51]. This is accomplished by so-called *learning algorithms* that compute the optimal weights and biases to solve a particular task. Today, the most widely-used learning algorithm in ANNs is back-propagation: briefly, it consists in propagating, at each learning iteration, the error on the training set *backwards* from the output to the input layers, in order to adjust the weights of the network[52].

Convolutional neural networks (CNNs) are a type of ANNs that contain *convolutional* layers, i.e., kernels with shared weights whose main task is to extract salient features from parts of the CNN input[53]. Sharing weights among neurons in the same convolutional layer makes training more efficient, thus allowing a substantial increase in the number of convolutional layers that can be "stacked" on top of each other, mimicking the high-level organization of the mammalian visual cortex[54]. This, together with the increase in computational power in recent years, has caused an exponential growth in the number of deep learning applications, in particular in tasks related to classification and data processing[52]. The architecture of the CNN used in this work is based on the `Deep Filtering` network introduced in ref. 35 to solve a similar parameter estimation problem as ours. As shown in Fig. S3, it contains three *preprocessing blocks*, each made up of one convolution and one max pooling layer. This preprocessing pipeline is replicated as many times as the number of electrical variables used as inputs to the CNN: for instance, if one were to use direct and quadrature voltages ($M = 2$) at two buses ($N = 2$), the total number of preprocessing pipelines would amount to four. Each of these pipelines receives as input 60 s of data sampled at 40 Hz, adding up to a total of 2400 samples. The outputs of the pipelines are concatenated and flattened into an array of $M \times N \times 64 \times 36 = 9216$ samples, which is fed to two fully-connected layers that perform the actual "classification", thus producing the estimated value of momentum. The loss function used during training is the conventional mean absolute error (MAE):

$$\text{MAE} = \frac{1}{n}\sum_{i=1}^{n}|y_i - x_i|, \tag{5}$$

where $n$ is the number of training traces and $y_i$ and $x_i$ are the predicted and true values of momentum, respectively. The size values indicated in Fig. S3 and used throughout this work refer to convolutional layers with 16, 32 and 64 neurons, kernel size and stride equal to 5 and 1, respectively, max pooling layers with pooling size of 4 and a first dense layer containing 64 neurons. The number of neurons in the second dense layer is equal to the number of predicted values of momentum, and thus $L = 1$ in this work, given that we want to estimate the momentum of only one area at a time. The nonlinear activation function in the densely connected layers is the rectified linear unit (ReLU) (not explicitly indicated in Fig. S3), while the preprocessing layers do not have a nonlinear activation function.

In the preliminary phases of this work we tested the effect on the accuracy of the CNN of various model hyperparameters, namely the size and stride of the kernels in the convolutional layers and the number of units in the max pooling layers. The results of this investigation, in the simple scenario of predicting only two momentum values, are summarized in Fig. S4 and Table S5: in the latter, the output size column indicates the number of outputs of the last max pooling layer, which corresponds to the number of inputs to the downstream fully connected layer. Each pair of convolutional and max pooling layers converts its $N_{in}$ input samples into $N_{out}$ output samples according to the following formula:

$$N_{out} = \left\lfloor \frac{\left\lfloor \frac{N_{in}-K_{sz}}{K_{str}} \right\rfloor + 1}{P_{sz}} \right\rfloor, \tag{6}$$

where $K_{sz}$ and $K_{str}$ are the kernel size and stride, respectively, and $P_{sz}$ is the number of units in the pooling layers. Applying this function recursively three times (i.e., the number of convolution/max pooling pairs in the preprocessing pipelines) with initial $N_{in} = 2400$ (i.e., the total number of samples used by the CNN), leads to the output size values indicated in Table S5. Notice that not all combinations of kernel size and stride and number of units in the pooling layers are feasible (i.e., they lead to $N_{out} = 0$ in Eq. (6)), hence the missing violin plots in the right panel of Fig. S4. Overall, we found that the combinations of 6 units in the max pooling layers, kernel stride equal to 1 and kernel sizes equal to either 3 or 5 gave the lowest validation error. However, we chose to use a set of hyperparameters that in our tests gave only marginally higher validation loss (median loss = 0.00386 vs. 0.00374): the reason for this lies in the fact that this particular set of hyperparameters leads to a higher number of output neurons from the preprocessing pipeline (36 instead of only 10) that then constitute the input to the downstream fully connected layer. We reasoned that this higher dimensionality of the input to the fully connected layer might endow it with superior generalization capabilities in more complicated tasks, such as those described in Figs. 5-7.

We used the Adam optimizer[55] either with a fixed learning rate of $5 \times 10^{-4}$ or with a cyclical learning rate[56] with initial and maximal learning rates of $5 \times 10^{-5}$ and $2 \times 10^{-3}$, respectively, and step size equal to 10 times the number of iterations in an epoch. Network weights were initialized using the Glorot approach as described in ref. 57.

All ML models were implemented in TensorFlow 2[58].

## Numerical simulations

Time-domain simulations were performed with the simulator PAN[59], exploiting its capability of properly solving numerically the stochastic differential equations that govern the considered power system model. To build the training set, as shown in Fig. 2, we sampled the $(H_{G_2}, H_{G_3})$ plane using a regularly spaced grid of $6 \times 6$ points: for each combination of inertia values we simulated the network for $300 \times 10^3$ s (i.e., approximately 83 h) and used these data to train the CNN. The data were normalized to have zero mean and unitary standard

deviation according to the formula

$$y_{ij} = \frac{x_{ij} - \mu_x}{\sigma_x}, \qquad (7)$$

where $x_{ij}$ is the $j$-th time sample of the $i$-th training trace, $\mu_x$ and $\sigma_x$ are the mean and standard deviation of $x$ over the whole training dataset and $y_{ij}$ are the corresponding normalized values. We chose this normalization to preserve the normal distribution of the voltage traces and to account for the range of voltage magnitudes at different buses in the IEEE 39-bus network. No other preprocessing (e.g., filtering) was applied to the data. The validation and test sets were built in a similar fashion, but with shorter simulations, as detailed in Table S6. Importantly, both the validation and test sets were also normalized according to Eq. (7) using the mean and standard deviation of the training set.

### Additional ML models

We used ML models available in Scikit-learn[60] as benchmarks for the performance of our CNN-based approach. For each of the following models, we indicate here only the hyperparameters whose values were different from Scikit-learn's implementation defaults. We did not investigate extensively whether other hyperparameter combinations led to better results, as this was outside the scope of this work.

- SVR[61]: Nu Support Vector Regression, with radial basis function (RBF) kernel, $\nu = 0.5$, and penalty parameter $C = 1$. The tolerance for stopping was set to $10^{-4}$.
- MLP: a densely-connected multi-layer perceptron with 1 hidden layer containing 67 units, leading to a number of trainable parameters comparable to that of the CNN. Training was performed over 2000 iterations, with early stopping if the loss function did not decrease for more than 200 iterations.
- K-nearest neighbors: $k = 5$ neighbors weighted by the inverse of their distance in the prediction phase (i.e., closer neighbors have a greater influence on the predicted value). The distance function was based on the Dynamic Time Warping similarity measure between time series[62], as implemented in the Python package tslearn[63].
- Kernel ridge[64]: ridge regression with an RBF kernel and regularization strength $\alpha = 0.1$.
- Random forest[65]: ensemble method based on fitting various binary decision tree regressors on the training data and then averaging their output in the prediction phase. The number of parameters of each tree depends chiefly on its depth, which is chosen automatically by the training algorithm if no maximum allowed value is specified. We used $N = 150$ decision trees to approximately match the total number of trainable parameters of the corresponding CNN, as shown in Table S4.

## Data availability

Additionally, the processed data have been deposited in Figshare under accession code (https://doi.org/10.6084/m9.figshare.23652730). Source data are provided with this paper.

## Code availability

The Python code used for *(i)* generating the synthetic data used in this paper, *(ii)* training the CNN and the other ML models, and *(iii)* generating Figs. 2-10 is available at Zenodo under accession code (https://doi.org/10.5281/zenodo.8123317) and at GitHub (https://github.com/danielelinaro/inertia).

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

## Acknowledgements

Work partially supported by Terna Rete Italia S.p.A., under research contract number ST236-DSC018.

## Author contributions

Conceptualization and methodology, D.L., F.B., and A.B.; investigation, D.L., F.B., D.d.G., and A.B.; formal analysis and software, D.L.; resources, C.P. and G.M.G.; writing—original draft, D.L.; writing—review and editing, all authors. Supervision, F.B., A.B., and S.G.

## Competing interests

The authors declare no competing interests.

## Additional information

**Peer review information** : *Nature Communications* thanks the anonymous reviewer(s) for their contribution to the peer review of this work. A peer review file is available.

