## [Peer Review File · Nature Communications]

Continuous estimation of power system inertia using convolutional neural networksREVIEWER COMMENTS

Reviewer #1 (Remarks to the Author):

The authors present a unique data-driven method to estimate power system inertia with an in-depth analysis of power spectra and input-to-output correlation analysis. It is indeed an interesting analysis to estimate inertia in a network comprising synchronous as well as non-synchronous units, which have been proven to change the system dynamics in an unusual way making the task of estimation even more difficult. Unlike other data-driven or machine-learning techniques, the authors have also provided insights into how the actual layers of the machine-learning models behave and what information can be obtained from the trained models. Such analysis is extremely relevant in critical infrastructure such as power systems, where the traditional machine learning (black box) model is not considered suitable due to a variety of reasons. The authors have also provided relevant literature on how the existing methods fail to work in a real system or how they are inappropriate in their methodology. The reviewer particularly appreciates the introduction of momentum, which brings a concept of incremental quantity into the system, unlike the center of inertia. The paper is well-written, with detailed descriptions of every concept required to understand the proposed work.

The reviewer has the following concerns regarding the proposed work. Some related questions are sub-numbered.

1.

- a. From the power spectral density in Fig. 2, the marked differences are not only in the frequency range of [0.5, 2] Hz but also in the range of [4, 10]Hz, which is not described well in the paper. What is causing this difference in the spectral density for the latter frequency range?
- b. A positive correlation value of less than 0.5 is often considered a poor correlation. How can the authors justify the correlation values presented in Fig. 3?

2.

- a. Why is the analysis conducted on the direct axis component of the voltage? Can the authors conduct an analysis of the relationship of momentum with the quadrature axis of the voltage?
- b. Is the analysis based on the fact that the transmission lines have a decoupled Pf and QV relationship? How does the analysis change when estimated at the point of common coupling where such relationships are no longer valid?

3. On Fig. 2. C, please discuss why the variance of the voltage distribution is not changing, but the mean is.

4. Please provide additional analysis on estimating the momentum of Area 2 with four generators and discuss what additional requirements and estimation complexity change as compared to the estimation of momentum in area 1.

5. In general, system inertia varies from 2s to 10s. Why did the authors select a specific range of 3s to 5s to train the neural network? Is the training data enough to generalize the CNN model?

6. The reviewer believes that damping is vital in decreasing the frequency nadir and thus supplementing the inertia in a system to minimize the rate of change of frequency. Thus, it will be interesting to know how the inclusion of damping in Eq. (1) can affect the estimation task.

7. The reviewer wants to highlight the following excerpts from the paper. There is a two-part question at the end:

- a. "These results indicate that, in order to achieve as accurate a prediction as possible, a CNN should be trained with data that cover as many spectral conditions as possible, as this is necessary to have an adequate tuning of the filters that constitute the preprocessing pipeline of the network."
- b. "These results once more highlight the capability of a CNN to correctly predict the area momentum in several operating scenarios, assuming that the

the network has been trained on an appropriately constructed dataset."

Q:

1. Isn't this true for any machine learning models? The more data, the better the results. How is the proposed method standing out as compared to other machine learning models if only the training-based method is compared (not comparing the input features or data extraction method)?
2. What if we add an additional compensator (may be two or four of them)? Do we need to train the CNN again? This seems a bit impractical for the reviewer.

8. Please explain what is meant by capable of correctly predicting in the following sentence:

"As a consequence, the CNN is capable of correctly predicting the value of area momentum only when the red and black PSDs overlap for frequencies < 1 Hz, as shown in Fig. 9B, where the dashed magenta traces are the correct values of momentum and the black trace is the moving average of the CNN prediction."

The inertia of VSGs are unknown. How does the proposed method estimate it?

9. Imagine energy storage units are in place with fast-frequency supports. What additional changes are required in the proposed approach to incorporate the effect of such IBR-based units?

Reviewer #2 (Remarks to the Author):

The manuscript proposes a convolution neural network (CNN)-based technique for continuous online inertia estimation in an electrical power system. The authors have demonstrated the power spectra analysis and input-output correlations while considering voltage measurements and stochastic load fluctuations. They have explained how the CNN operates and needs training in specific operating condition in their proposed method. Moreover, the authors have validated the applicability of their proposed method using synthetic data generated by simulating the IEEE 39-bus benchmark system. The proposed method was able to determine the effective frequency band range that carried the most input feature information for proper CNN training.

The manuscript has relevant contributions to the field of power systems, however the reviewer has some concerns and recommendations for the manuscript.

Major Comments:

- Although both the CNN and Artificial Neural Network (ANN) are neural networks, they differ in their architecture and purpose. The manuscript uses both ANN and CNN interchangeably, which are the foundation of the manuscript, but they are not explained clearly. As a result, readers may have difficulty understanding the concepts and research findings. To address this issue, it is recommended to add more detailed explanation of the theory of ANN and CNN, supported by relevant references and graphics. This will enhance readers' understanding of the concepts.
- The authors discuss the state-of-the-art in inertia estimation and suggest that the accuracy of different approaches is dependent on the type of input used to train the model. They presented their study as a simple yet accurate method using ambient measurements, but there is no comparison made to literature to support the claims made. For example, active perturbation methods are dismissed in a single sentence, yet have been successfully deployed in real power systems, e.g., J. Pierre et al., "Probing Signal Design for Power System Identification," IEEE Trans. Pow. Sys., vol. 25, 2010 (actively perturbed WECC); or other ambient measurement-based approaches, e.g., D. Yang et al., "Ambient-Data-Driven Modal-Identification-Based Approach to Estimate the Inertia of an Interconnected Power System," IEEE Access, vol. 8, 2020. The authors need to support their claims of superior model accuracy and other mentioned limitations (e.g., calibration, computational requirements, effect of measurement accuracy) compared to other literature approaches.

- In Line 519, "...only to the [1.5; 3] Hz frequency range...". Here, from Fig. 5 C-D, the sensitive range seems to be [0.5, 1.3] Hz. However, the authors mentioned the sensitive range as [1.5; 3] Hz. The author should clarify the statement or figure.
- The analysis of the frequency spectrum is performed up to 20 Hz, however there was no justification for this range. It is suggested the authors include the explanation of the minimum and maximum limit for the frequency range [0.1, 20] Hz of the study.
- The manuscript discusses on the online estimation of momentum considering the spectral analysis of the measured voltage with the stochastic load change and input-output correlations of the convolutional layer. However, they do not clearly explain how the measured voltage with respect to the stochastic load change are correlated, and hence how they effect the estimation of the continuous momentum. The authors also do not show what the generated synthetic load they used for the analysis. It is recommended to add a voltage profile with respect to load change to the manuscript with additional discussion.
- The accuracy of the convolution layer depends on the choice of window size and stride, which decides the area size and information to capture. E.g., larger window size and smaller stride increases the required amount of computation and impacts accuracy, and vice versa. The authors did not explain how they subdivided this network to construct the layers. It is recommended to add a more detailed explanation of the process of subdividing the layers.
- The proposed CNN approach appears to need tuning and training with the introduction of each system change, which limits the applicability of the approach. Moreover, the selection process of weight initialization for the CNN was not clearly discussed. The authors are using the each system output response to adjust the model, which seems to contradict the claim of not needing the system information or updating the output. It is recommended to clarify the contribution of the proposed method to the literature if these claims are not correct, or if the reviewer misinterpreted then the CNN online tuning/training needs to be clarified.
- The authors claim that the trained CNN model on the voltage recorded at the bus is capable of correctly estimating the area momentum, but only with the tuning of filters. For example, in Line 440, the author has mention that "...and it does so by tuning in its preprocessing pipeline". However, the manuscript does not mention clearly the input data is preprocessed, e.g., outliers, missing data, etc. It is suggested the authors add a clear description on selection and tuning of the filters, including the data preprocessing pipeline.
- In Line 767, "... the optimization algorithm during training...", the manuscript has mentioned the use of an optimization algorithm for determining the frequency band, however it is not clear from the explanation what optimization was used. Please clarify.

Minor Comments:

- Abbreviations should be defined in the abstract and once with the first use in the context.
 - o Abbreviated form is used but not defined. For instance, SEM should be defined in the text. It is suggested the authors to maintain the consistency.
- Authors should maintain consistency while using terms like "Figure", "Figs.", and "Fig.".
- Figures should be of higher quality, as well as addition of legends in the figures should be more appropriate. Moreover, there is no figure in Fig. 5, subplot A.
- The range of frequency values is confusing against the reference format. E.g., on Line 359 the range [1,20] could be confused with citing references [1] and [20].

Response to referees for the paper
*Continuous Estimation of Power System
Inertia Using Convolutional Neural
Networks*

Daniele Linaro*, Federico Bizzarri, Davide del
Giudice, Cosimo Pisani, Giorgio M. Giannuzzi, Samuele
Grillo and Angelo Brambilla

*Corresponding author(s). E-mail(s): daniele.linaro@polimi.it;

Response to reviewer 1

The authors present a unique data-driven method to estimate power system inertia with an in-depth analysis of power spectra and input-to-output correlation analysis. It is indeed an interesting analysis to estimate inertia in a network comprising synchronous as well as non-synchronous units, which have been proven to change the system dynamics in an unusual way making the task of estimation even more difficult. Unlike other data-driven or machine-learning techniques, the authors have also provided insights into how the actual layers of the machine-learning models behave and what information can be obtained from the trained models. Such analysis is extremely relevant in critical infrastructure such as power systems, where the traditional machine learning (black box) model is not considered suitable due to a variety of reasons. The authors have also provided relevant literature on how the existing methods fail to work in a real system or how they are inappropriate in their methodology. The reviewer particularly appreciates the introduction of momentum, which brings a concept of incremental quantity into the system, unlike the center of inertia. The paper is well-written, with detailed descriptions of every concept required to understand the proposed work.

We would like to thank the reviewer for the careful assessment of our work.

The reviewer has the following concerns regarding the proposed work. Some related questions are sub-numbered.

1. a *From the power spectral density in Fig. 2, the marked differences are not only in the frequency range of $[0.5, 2]$ Hz but also in the range of $[4, 10]$ Hz, which is not described well in the paper. What is causing this difference in the spectral density for the latter frequency range?*

Answer: synchronous generators are electro-mechanical elements characterized by frequency footprints (see, for instance, the peaks in Fig. 2) in different frequency bands. Peaks in the $(4, 10)$ Hz frequency range are mostly due to inter-area oscillations and display low sensitivity with respect to the inertia constant of the synchronous generators. As a consequence, all the PSDs in Fig. 2 overlap in this frequency range because we changed only the generators' inertia constants for the generation of the data in that figure. On the other hand, in Fig. 5 we see significant differences in the high-frequency peaks, where also the inertia of a compensator was changed. Indeed, compensators can cause inter-area oscillations that can be exploited to extrapolate their inertia contribution [1].

We have added the following paragraph at the end of the section "Voltage spectra upon area momentum variation" to clarify this point:

"For instance, one can clearly see that peaks in the $(4, 10)$ Hz frequency band in Figs 2D,E do not change substantially when the inertia of G2 and G3 is varied: indeed, these peaks are due to inter-area oscillation modes [1] and as will be shown in the following they are mainly due to the inertia of synchronous compensators."

- b *A positive correlation value of less than 0.5 is often considered a poor correlation. How can the authors justify the correlation values presented in Fig. 3?*

Answer: this is an interesting point raised by the reviewer. While we agree that, in general, a correlation slightly above 0.4 as the one shown in Fig. 3 would not be considered a “strong correlation”, several factors give us confidence that what we are seeing in our analysis is indeed a reasonable explanation for the way the CNN operates. First of all, the correlation values shown in Fig. 3 are in line with those in the paper that introduced this type of analysis [2]: we do show slightly lower values, but this is probably attributable to the clear and functionally-relevant frequency structure observed in EEG data like those studied in [2]. Secondly, the magnitude of the (significant) correlations in our analysis can be increased by reducing the number of subdivisions of the frequency interval (0.1, 20) Hz, as shown in Fig. S1. Third, what we see are not individual values of correlations, but rather an organized structure of the correlation landscape that is unlikely to be due to chance. Finally, by computing the difference between the correlation values obtained with the trained and untrained network, we further strengthen the relevance of the analysis. Concerning the new supplementary figure, we have added references to it in several sections of the manuscript.

2. a *Why is the analysis conducted on the direct axis component of the voltage? Can the authors conduct an analysis of the relationship of momentum with the quadrature axis of the voltage?*

Answer: we have chosen to use the direct axis component at the voltage at bus 3 because in our preliminary analyses it showed a clear dependence on the momentum of the area, both in terms of distribution and spectrum, as shown in Fig. 2C-E. However, this happens to be the case also for the quadrature axis component of the voltage at the same bus, as we now show in Fig. S2B. In other words, we performed a rough sensitivity analysis of the voltages at several buses to identify those that carried the most information: however, defining a rigorous methodology to perform this type of sensitivity analysis, albeit interesting and potentially extremely useful, is beyond the scope of this work. We have updated the manuscript to address the reviewer’s concern by stating that “*Similar results can be obtained by training a CNN using $V_{q,3}$, i.e., the quadrature axis component of the voltage at bus 3, as shown in Fig. S2B.*”

- b *Is the analysis based on the fact that the transmission lines have a decoupled Pf and QV relationship? How does the analysis change when estimated at the point of common coupling where such relationships are no longer valid?*

Answer: traditionally, the direct axis component is associated to the active power transfer (assuming a high power factor). When applied to a system where P/f and Volt/Var regulations are decoupled, this could

4 *Inertia estimation with CNNs*

lead to assuming that only the direct axis is relevant when focusing on frequency variations. However, as stated early, if one relies on the statistical properties of measurements time series, this simplifying assumption is not relevant and the most information carrying measurements should be employed for training the CNN. Indeed, as described in the answer to the previous question (i.e., 2a), the CNN can employ as input direct or quadrature voltage measurements without any significant variation in momentum estimation accuracy, even though a transmission system is used as test benchmark.

3. *On Fig. 2. C, please discuss why the variance of the voltage distribution is not changing, but the mean is.*

Answer: we thank the reviewer for pointing out this unclear aspect of our presentation. Indeed, the standard deviation of the distributions shown in Fig. 2C changes as a function of area momentum, as we now show in the inset of Fig. 2C. The mean, on the other hand, is the same for all distributions and is (approximately) equal to 0. This is due to the fact that the whole dataset is normalized to have zero mean and unitary standard deviation (as we now explain in the Methods): since the operating point of the power system does not change when inertia is varied, the mean of the voltage traces for all momentum values is the same. The standard deviation of the voltage traces, on the other hand, changes with the inertia of the generators, leading to the different distributions observed in Fig. 2C. To better explain these aspects, we have added the following paragraph to the section “Voltage spectra upon area momentum variation”:

“While the effect of changing the area momentum is not evident on the example voltage traces shown in Fig. 2B (which are normalized, over the whole dataset, to have zero mean and unitary standard deviation, see Methods), it becomes more apparent when looking at the shape of the distributions of the voltage samples over longer simulation times, as shown in Fig. 2C. Indeed, the distributions’ means are approximately 0 for all momentum values: this is a consequence of having subtracted, when normalizing, the voltage value of the power-flow (PF) solution, which is not affected by the inertia of the generators. The standard deviation of the distributions, on the other hand, is affected by the values of inertia, and increases with the overall area momentum, as shown in the inset of Fig. 2C”.

4. *Please provide additional analysis on estimating the momentum of Area 2 with four generators and discuss what additional requirements and estimation complexity change as compared to the estimation of momentum in area 1.*

Answer: To address this question, we have repeated the analysis in Fig. 3 of the manuscript for area 2, limited to the case of high and low area momentum. As expected, a CNN trained on this new data is capable of correctly estimating the momentum values, albeit with higher MAPE than what we observed for area 1. We have summarized this analysis in Fig. S2,

and added the following sentence to the manuscript:

“As shown in Fig. S2, this correlation analysis was carried out on CNNs trained to predict the momentum values of either area 1 or area 2 using either the direct or quadrature axis components of the voltages at bus 3. In all cases, the CNN was capable of correctly predicting the momentum—albeit with significantly higher MAPEs in the case of area 2, see Fig. S2C,D—, and with correlation maps characterized by strikingly similar structures in all cases.”

5. *In general, system inertia varies from 2s to 10s. Why did the authors select a specific range of 3s to 5s to train the neural network? Is the training data enough to generalize the CNN model?*

Answer: we used an interval of $(-1, 1)$ s around the default values of inertia of the IEEE 39-bus network. While this range can of course be expanded (potentially using a coarser grid to avoid an extremely large training dataset), it was out of the scope of this work to perform an exhaustive sampling of all the possible combinations of inertia values that would be observed in a real-world scenario. Concerning the generalization capabilities of the CNN to momentum values outside of those considered during training, we have generated a test set spanning the grid of inertia values $H_1 \times H_2 = (2, 10)$ s \times $(2, 10)$ s and used the CNN trained on the full training set shown in Fig. 2A to predict the area momentum. The results of this experiment are shown in Fig. R1: as it can be seen, once the values of inertia are too far from those contained in the original training set (grey shaded area in Fig. R1), the MAPE of the prediction increases substantially. However, this is not unexpected as most ANN display poor generalization capabilities once their inputs are too far off from the data they were trained on. We have decided not to include this additional figure in the new version of the manuscript (or in the Supplementary Materials), as we feel that it might convey a misleading message to the reader: however, if the reviewer feels that the addition of these results is necessary, we’d be happy to oblige.

6. *The reviewer believes that damping is vital in decreasing the frequency nadir and thus supplementing the inertia in a system to minimize the rate of change of frequency. Thus, it will be interesting to know how the inclusion of damping in Eq. (1) can affect the estimation task.*

Answer: we agree with the reviewer that load damping is a crucial parameter influencing the stability of a power network. To address this specific comment, we have performed additional simulations where we used a CNN to predict the values of momentum of area 1 when the load damping of generators G2 and G3 is varied, while their inertia is kept constant. The results are shown in Fig. 10: interestingly, the effect of load damping on the PSDs appears either to be limited to very low frequencies (i.e., up to approximately 0.1 Hz) or to affect only the amplitude of one of the peaks. As a consequence, load damping has only a minor influence on the MAPE of the CNN prediction, which, in this particular case, ranges from approximately

Fig. R1 MAPE on an extended test set: each dot represents a pair (H_{G2}, H_{G3}) used to generate $N = 150$ 1 min-long traces, which were subsequently used as input to a CNN trained on the dataset shown in Fig. 2A of the manuscript (corresponding to the gray shaded area in this figure). The size and color of each marker are proportional to the MAPE, according to the legend and colorbar shown.

1.8% for the case with no load damping (corresponding to the condition in which the CNN was trained) to approximately 3.4% when the value of load damping is set to 4. We have added the subsection titled “Robustness to load damping variability” to the discussion to describe these findings and added a load damping term to Eqs. 1 and 2.

7. *The reviewer wants to highlight the following excerpts from the paper. There is a two-part question at the end:*

“These results indicate that, in order to achieve as accurate a prediction as possible, a CNN should be trained with data that cover as many spectral conditions as possible, as this is necessary to have an adequate tuning of the filters that constitute the preprocessing pipeline of the network.”

“These results once more highlight the capability of a CNN to correctly predict the area momentum in several operating scenarios, assuming that the network has been trained on an appropriately constructed dataset.”

- a *Isn't this true for any machine learning models? The more data, the better the results. How is the proposed method standing out as compared to other machine learning models if only the training-based method is compared (not comparing the input features or data extraction method)?*

Answer: while we agree with the reviewer that typically more training data leads to better prediction results, this is not true in every possible case and scenario, especially if additional training data does not equate to a better sampling of parameter space. To address the second part of the reviewer’s question, we have now compared our CNN-based approach

to several other established ML models used for regression and find that our method outperforms all of them. The results of this analysis have been added to the discussion, in the subsection entitled “Comparison with other ML methods”.

- b *What if we add an additional compensator (may be two or four of them)? Do we need to train the CNN again? This seems a bit impractical for the reviewer.*

Answer: that depends on how the additional compensator(s) might affect the spectral footprint of the bus voltages used to estimate the value of momentum. The best-case scenario is that, once the CNN has reached a certain level of training, it gets better at generalizing and therefore there is no need to further add training data. More likely, one will have to train a new CNN to address the presence of one or more new compensators.

8. *Please explain what is meant by capable of correctly predicting in the following sentence:*

“As a consequence, the CNN is capable of correctly predicting the value of area momentum only when the red and black PSDs overlap for frequencies < 1 Hz, as shown in Fig. 9B, where the dashed magenta traces are the correct values of momentum and the black trace is the moving average of the CNN prediction.”

The inertia of VSGs are unknown. How does the proposed method estimate it?

Answer: the model of VSGs used in this work includes as a parameter the virtual inertia that these elements synthesize through an *ad hoc* control strategy aimed at mimicking the inertia constant of a traditional synchronous machine. As a result, the inertia of VSGs can be easily modified during simulation to train the proposed CNN and assess its accuracy.

For the analysis in Fig. 9, we have substituted the synchronous generator G3 with a corresponding VSG providing synthetic inertia. Therefore, it is “as if” the momentum of area 1 increased when the virtual inertia provided by the VSG is changed and that is what the network estimates.

In real power systems, the inertia of VSGs is not a directly accessible parameter. This implies that whoever is in charge of training the CNN to estimate the overall power system momentum would require some methodology to infer the virtual inertia of VSGs while creating the training set. This problem, however, is outside the scope of this work.

9. *Imagine energy storage units are in place with fast-frequency supports. What additional changes are required in the proposed approach to incorporate the effect of such IBR-based units?*

Answer: we distinguish between two main ancillary services an IBR-based unit contributes:

- a grid-forming IBR emulates the swing equation of the synchronous generator, i.e., its mechanical behavior. This does not mean that a grid-forming

IBR replicates the full dynamics of the synchronous generator, in fact its electrical part is not emulated. An energy storage system connected to the grid by a grid-forming IBR contributes inertia; it can behave just like a synchronous compensator with average zero power exchange, losses excluded.

- A grid-following IBR follows the frequency behavior at the point of common coupling by using a phase-locked loop that closely and as accurately as possible tracks the electrical phase/frequency of the bus voltage. The grid-following IBR implements a dynamic frequency-dependent load that contributes damping but not inertia. An energy storage system connected to the grid by a grid-following IBR contributes load damping, just like a dynamic frequency dependent load.

In our work, we trained the CNN to estimate only the momentum and not the damping.

Response to Reviewer 2

The manuscript proposes a convolution neural network (CNN)-based technique for continuous online inertia estimation in an electrical power system. The authors have demonstrated the power spectra analysis and input-output correlations while considering voltage measurements and stochastic load fluctuations. They have explained how the CNN operates and needs training in specific operating condition in their proposed method. Moreover, the authors have validated the applicability of their proposed method using synthetic data generated by simulating the IEEE 39-bus benchmark system. The proposed method was able to determine the effective frequency band range that carried the most input feature information for proper CNN training.

The manuscript has relevant contributions to the field of power systems, however the reviewer has some concerns and recommendations for the manuscript.

We would like to thank the reviewer for the careful assessment of our work.

Major Comments

- *Although both the CNN and Artificial Neural Network (ANN) are neural networks, they differ in their architecture and purpose. The manuscript uses both ANN and CNN interchangeably, which are the foundation of the manuscript, but they are not explained clearly. As a result, readers may have difficulty understanding the concepts and research findings. To address this issue, it is recommended to add more detailed explanation of the theory of ANN and CNN, supported by relevant references and graphics. This will enhance readers' understanding of the concepts.*

Answer: we agree with the reviewer that we have used “ANN” and “CNN” somewhat interchangeably in a way that might confuse a reader not fully acquainted with the terminology. We have fixed the problematic instances

throughout the manuscript and added the following paragraph in the Introduction to better clarify the distinction between the two:

“In this paper, we develop a framework for the online estimation of momentum based on a CNN [21,22]: CNNs are a particular category of ANNs [23] that have become the de facto standard for classification tasks [24], particularly in the field of computer vision [25]. The inputs to the CNN are the time series of a set of electrical quantities recorded at a reduced number of buses of the network, while the output is the momentum of one or more areas of the power network. The goal of our approach is therefore to have a CNN learn the relationship between time series of electrical variables and corresponding values of momentum.”

Additionally, we have extended the “Convolutional neural network architecture” section of the Methods to give some background on ANNs and CNNs together with appropriate references.

- *The authors discuss the state-of-the-art in inertia estimation and suggest that the accuracy of different approaches is dependent on the type of input used to train the model. They presented their study as a simple yet accurate method using ambient measurements, but there is no comparison made to literature to support the claims made. For example, active perturbation methods are dismissed in a single sentence, yet have been successfully deployed in real power systems, e.g., J. Pierre et al., “Probing Signal Design for Power System Identification,” *IEEE Trans. Pow. Sys.*, vol. 25, 2010 (actively perturbed WECC); or other ambient measurement-based approaches, e.g., D. Yang et al., “Ambient-Data-Driven Modal-Identification-Based Approach to Estimate the Inertia of an Interconnected Power System,” *IEEE Access*, vol. 8, 2020. The authors need to support their claims of superior model accuracy and other mentioned limitations (e.g., calibration, computational requirements, effect of measurement accuracy) compared to other literature approaches.*

Answer: We do not fully agree with the reviewer’s comment that our manuscript does not present comparisons to other methods available in the literature: as a matter of fact, in the original version of the manuscript we already provided the reader with some comparisons with other ML-based approaches to the problem of inertia estimation in power systems. In particular, we wrote in the discussion: *“Previous research has investigated the application of ANNs [38,39] in general and CNNs in particular [40] to the problem of inertia estimation. While similar in the overall approach and accuracy of the prediction, our method presents two clear advantages: first, it is entirely data-driven, as it relies exclusively on the voltage fluctuations attributable to the stochasticity of power loads. This is not the case with other methods, such as [40], which instead require probing signals to perturb the system from its steady state operation point. Secondly, for the first time we provide an in-depth analysis of the mechanisms underlying the functioning of the CNN, thus providing crucial guidelines for the choice of the most*

appropriate training set to achieve a desired level of accuracy in the prediction of network momentum.” In the new version of the manuscript, this part is in a subsection of the discussion titled “Comparison with other ML methods”, to better highlight it within the context of the discussion.

Concerning the comparison with other methods, like the one cited by the reviewer, the literature on inertia estimation in power systems is vast and has been growing at a quickening pace. For instance, in Elsevier’s abstract and citation database Scopus, papers including the keywords “inertia” and “power system” in their abstract were only 30 in the year 2000, while this figure spiked to 1200 in 2022.

An exhaustive comparison with the methods that are already available becomes thus more and more difficult, especially because codes and procedures developed by researchers are rarely shared on the web. To reliably re-implement algorithms and run test cases from other papers is a prohibitive task since usually the exact parameter values of the systems under test are not available. Furthermore, many technicalities that make a code efficient are often omitted. This is one of the reasons why we decided to foster the reproducibility of our results by sharing our work on GitHub.

All the results presented in this manuscript come with the corresponding MAPE on the prediction, which rarely exceeded 4%. This allows the reader to quantitatively appreciate the accuracy of the proposed method, and make his/her own comparisons with other methods, for instance with the recent ones based on ANNs that we already mentioned in the discussion of the manuscript.

As far as the references suggested by the Reviewer are concerned, J. Pierre et al., “Probing Signal Design for Power System Identification,” IEEE Trans. Pow. Sys., vol. 25, 2010, specifically studies low-level probing signal design for power-system identification. We are aware of the fact that inertia estimation can be viewed as a specific task of power-system identification, but this very paper does not even mention this aspect. We added a citation to the paper by D. Yang et al., “Ambient-Data-Driven Modal-Identification-Based Approach to Estimate the Inertia of an Interconnected Power System”, in addition to citing in the introduction two more extensive review papers that provide the reader with a more exhaustive description of the scientific landscape [3, 4].

- *In Line 519, “. . . only to the [1.5; 3] Hz frequency range . . . ”. Here, from Fig. 5 C-D, the sensitive range seems to be [0.5, 1.3] Hz. However, the authors mentioned the sensitive range as [1.5; 3] Hz. The author should clarify the statement or figure.*

Answer: the range mentioned by the reviewer refers to the frequency band with the strongest inputs and hence the larger (spurious) values of correlation. We have clarified this point by adding the following paragraph in the results:

“As discussed for Fig. 3, high correlation values in the range (0.5, 1) Hz are due to the strong signal components that are present in the spectra for all

values of momentum: these reliably drive the output of the preprocessing pipelines even in the case of the untrained network (see Fig. 3G) and are therefore not used by the CNN to perform the classification.”

- *The analysis of the frequency spectrum is performed up to 20 Hz, however there was no justification for this range. It is suggested the authors include the explanation of the minimum and maximum limit for the frequency range [0.1, 20] Hz of the study.*

Answer: the reason for this choice lies in the fact that electrical power systems are characterized by electro-mechanical modes whose frequency content is limited to 20 Hz (at most). For instance, the frequency footprint of synchronous machines and motors and inverter-based resources contributing virtual inertia are all within this frequency band and therefore considering a maximum frequency above 20 Hz would lead to no additional information. We have added the following sentence in the text to clarify this aspect:

“The choice of this frequency band is dictated by the location in the frequency domain of the electro-mechanical modes of an electrical power system that display sensitivity to the inertia of synchronous machines/motors and of the virtual inertia provided by inverter-based resources: indeed, these are all located well within the 20 Hz upper frequency limit we have chosen.”

- *The manuscript discusses on the online estimation of momentum considering the spectral analysis of the measured voltage with the stochastic load change and input-output correlations of the convolutional layer. However, they do not clearly explain how the measured voltage with respect to the stochastic load change are correlated, and hence how they effect the estimation of the continuous momentum. The authors also do not show what the generated synthetic load they used for the analysis. It is recommended to add a voltage profile with respect to load change to the manuscript with additional discussion.*

Answer: we agree with the reviewer that, in the original version of the manuscript, the “Stochastic loads” section of the Methods did not provide enough information. We extended this section in the revised version to better explain how the measured voltages are related to the stochastic fluctuation of the loads. To avoid worsening the manuscript’s readability, mathematical details were not explicitly provided. Nevertheless, proper references were given in order to guide the interested reader into a deeper analysis of this aspect. Concerning the latter part of the reviewer’s question, it is our opinion that a plot representing the voltage profile with respect to load change can not be easily obtained because the IEEE 39-bus network contains 19 loads that fluctuate simultaneously and independently. The explanation that was given in terms of stochastic differential equations theory should be sufficient to support our claims.

- *The accuracy of the convolution layer depends on the choice of window size and stride, which decides the area size and information to capture. E.g.,*

larger window size and smaller stride increases the required amount of computation and impacts accuracy, and vice versa. The authors did not explain how they subdivided this network to construct the layers. It is recommended to add a more detailed explanation of the process of subdividing the layers.

Answer: as we now better explain in the Methods, we have tested various kernel sizes and strides for the convolutional layers, in addition to the number of units in the max pooling layers. As shown in the new Fig. S3 and Table S1, we found that the combinations of 6 units in the max pooling layers, kernel stride equal to 1 and kernel sizes equal to either 3 or 5 give the lowest validation error. However, we chose to use a set of hyperparameters that in our tests gave only marginally higher validation loss (median: 0.00386 vs. 0.00374) because this set leads to a higher number of output neurons from the preprocessing pipeline (36 instead of only 10) that then constitute the input to the downstream fully connected layer. We have modified the Methods section to better explain the reasoning behind our choices and to describe the related supplementary materials.

- *The proposed CNN approach appears to need tuning and training with the introduction of each system change, which limits the applicability of the approach. Moreover, the selection process of weight initialization for the CNN was not clearly discussed. The authors are using the each system output response to adjust the model, which seems to contradict the claim of not needing the system information or updating the output. It is recommended to clarify the contribution of the proposed method to the literature if these claims are not correct, or if the reviewer misinterpreted then the CNN online tuning/training needs to be clarified.*

Answer: we agree with the reviewer that our approach requires that the CNN be retrained when a new device is added to the network. This however happens only when the new device (*i*) provides inertia to the network, and (*ii*) adds its own specific signature to the PSD of the power system: thus, while we are not claiming that a once-trained CNN will correctly estimate inertia forever, we also do not think that network-changing events such as the ones outlined above will happen frequently in a real-world scenario. Additionally, we have decided to incrementally add new devices to our test system (i.e., first a compensator and then a VSG) specifically to highlight the relative ease with which one could tackle such power network changes. If we had limited ourselves to a fixed network structure, like most other works in the literature do, there would have been no need for retraining the CNN. Concerning weight initialization, we have added the following sentence to the Methods to explain how initialization is performed: “*Network weights were initialized using the Glorot approach as described in [57]*”.

- *The authors claim that the trained CNN model on the voltage recorded at the bus is capable of correctly estimating the area momentum, but only with the tuning of filters. For example, in Line 440, the author has mention that “...and it does so by tuning in its preprocessing pipeline”. However, the*

manuscript does not mention clearly the input data is preprocessed, e.g., outliers, missing data, etc. It is suggested the authors add a clear description on selection and tuning of the filters, including the data preprocessing pipeline.

Answer: we thank the reviewer for pointing out this missing piece of information. As now explained in the “Numerical simulations” subsection of the Methods, the voltage traces in the training set were normalized to have overall zero mean and unitary standard deviation. The validation and training sets, on the other hand, were normalized using mean and standard deviation of the training set. No other preprocessing (e.g., filtering) was applied to the simulated voltage traces.

- *In Line 767, “...the optimization algorithm during training ...”, the manuscript has mentioned the use of an optimization algorithm for determining the frequency band, however it is not clear from the explanation what optimization was used. Please clarify.*

Answer: we thank the reviewer for pointing out this missing piece of information in the Methods. We have used the Adam optimizer throughout our work, as now indicated with the following sentence:

“We used the Adam optimizer [55] either with a fixed learning rate of 5×10^{-4} or with a cyclical learning rate [56] with initial and maximal learning rates of 5×10^{-5} and 2×10^{-3} , respectively, and step size equal to 10 times the number of iterations in an epoch.”

Minor Comments

- *Abbreviations should be defined in the abstract and once with the first use in the context.*

Answer: we have defined the missing abbreviations at the first occurrence.

- *Abbreviated form is used but not defined. For instance, SEM should be defined in the text. It is suggested the authors to maintain the consistency.*

Answer: we have defined the missing abbreviation at the first occurrence.

- *Authors should maintain consistency while using terms like “Figure”, “Figs.”, and “Fig.”.*

Answer: we have checked that we use these terms consistently. “Figure” or “Figures” appear at the beginning of a sentence, whereas “Fig.” and “Figs.” are abbreviated forms used in the middle of a sentence.

- *Figures should be of higher quality, as well as addition of legends in the figures should be more appropriate. Moreover, there is no figure in Fig. 5, subplot A.*

Answer: we have made some minor improvements to our figures, as we believe that their level of clarity and quality matched the standard required by Springer. Concerning the specific subplot the reviewer refers to, we were

not able to understand this comment: if he/she refers to the blue trace in Fig. 5A, which is not easily distinguishable, we have improved the caption to clarify this point.

- *The range of frequency values is confusing against the reference format. E.g., on Line 359 the range [1,20] could be confused with citing references [1] and [20].*

Answer: we now use round brackets to indicate ranges, instead of square ones like in the previous version of the manuscript.

References

- [1] Hadavi, S., Phu Me, S., Bahrani, B., Fard, M., Zadeh, A.: Virtual synchronous generator versus synchronous condensers: an electromagnetic transient simulation-based comparison. *CIGRE Science and Engineering* **2022**(24) (2022). Publisher Copyright: © 2022- CIGRE.
- [2] Schirrmeister, R.T., Springenberg, J.T., Fiederer, L.D.J., Glasstetter, M., Eggenberger, K., Tangermann, M., Hutter, F., Burgard, W., Ball, T.: Deep learning with convolutional neural networks for EEG decoding and visualization. *Human brain mapping* **38**(11), 5391–5420 (2017)
- [3] Kontis, E.O., Pasiopoulou, I.D., Kirykos, D.A., Papadopoulos, T.A., Papa-georgiou, G.K.: Estimation of power system inertia: A comparative assessment of measurement-based techniques. *Electric Power Systems Research* **196**, 107250 (2021)
- [4] Prabhakar, K., Jain, S.K., Padhy, P.K.: Inertia estimation in modern power system: A comprehensive review. *Electric Power Systems Research* **211**, 108222 (2022). <https://doi.org/10.1016/j.epsr.2022.108222>

REVIEWERS' COMMENTS

Reviewer #1 (Remarks to the Author):

The reviewer would like to thank the authors for a comprehensive rebuttal. This is excellent work and timely when power system operators continuously refrain from using machine learning on AI-based methods. This work uniquely uses CNN to estimate the system's inertia without system perturbation, and the contribution convinces the reviewer.

The reviewer has no further comments as the authors have answered all the questions.

Reviewer #2 (Remarks to the Author):

The manuscript's authors have carefully addressed all of my comments and suggestions during the revision process. The revised manuscript is of high quality.